# GRAFT: Gradient-Aware Fast MaxVol Technique for Dynamic Data Sampling

## Abstract

Training modern neural networks on large datasets is computationally and environmentally costly. We introduce GRAFT, a scalable in-training subset selection method that (i) extracts a low-rank feature representation of the training data, (ii) applies a Fast MaxVol sampler to pick a small, diverse subset that spans the data's dominant subspace, and (iii) dynamically adjusts the subset size using a gradient-approximation criterion. Because selection operates inside the optimizer's mini-batches, its cost is independent of dataset size, and the same volume-maximizing criterion extends without change to one-shot curation of a full data pool, including instruction-tuning corpora for language models. By operating in low-rank subspaces and training on carefully chosen examples instead of full batches, GRAFT preserves the training trajectory while reducing wall-clock time, energy, and $CO_2$ emissions. Across CIFAR-10/100, FashionMNIST, TinyImageNet, Caltech256, and ImageNet-1K (ResNet-18), GRAFT lies on the Pareto frontier of accuracy and emissions against standard integrated subset-selection methods, and unlike them it carries over to language-model fine-tuning; its selection criterion further extends, without labels or gradients, to instruction-data curation, where we give a preliminary demonstration that it selects more diverse subsets than the usual baselines. At matched accuracy it consistently lowers $CO_2$, and its margin is largest in the aggressive-pruning regime (5 to 25% of the data), where principled selection matters most. On ImageNet-1K, GRAFT comes within 0.4 points of full-data ResNet-18 accuracy (70.0% against 70.4%) using 30% of the data and under a third of the emissions. We also delimit where GRAFT does *not* help, namely near-full data fractions and highly imbalanced batches, so practitioners can predict when the selection overhead pays off. An open-source implementation will be released to support reproducibility.

## 1 Introduction

Deep neural networks are powerful but costly to train, requiring substantial computational resources and energy, raising scalability and environmental concerns. A key driver of this cost is the large volume of training data, which increases optimization steps, memory use, and training time. Data subset selection offers a solution by using a smaller, well-chosen subset to maintain training efficiency while reducing redundancy. This can speed up training, cut energy use, and aid resource-constrained settings. However, many existing methods have drawbacks, such as costly preprocessing, reliance on proxy models, architecture restrictions, or repeated metric evaluations, limiting their flexibility.

In this work, we introduce GRAFT (Gradient-Aware Fast MaxVol Technique), a scalable and lightweight framework designed to integrate subset selection directly into the training loop. GRAFT operates in two main stages. First, it extracts compact feature embeddings using a low-rank projection. This step maps the high-dimensional representation of the data onto a lower-dimensional subspace, capturing the most salient features with minimal redundancy. Next, GRAFT performs a fast MaxVol sampling to select a subset of examples that best span the dominant subspace of the projected features. Unlike heuristic-based selection strategies, the MaxVol criterion guarantees that the chosen subset retains maximal expressiveness in terms of subspace coverage. To maintain alignment with the training dynamics, GRAFT introduces an adaptive sampling mechanism that dynamically adjusts the number of selected samples. This adjustment is guided by

measuring the alignment between the gradient computed over the selected subset and the gradient of the full batch. GRAFT evaluates the subspace representation of the selected subset to keep it representative of the learning dynamics. When the selected subset fails to span the full-batch gradient within a set tolerance GRAFT enlarges it, and when it spans the gradient comfortably GRAFT shrinks it to save computation. Crucially, this alignment check only *sizes* the subset; it never drives the selection itself.

This is what sets GRAFT apart from the dominant family of integrated selectors, which are built on *gradient matching*. Those methods cast selection as an optimization that repeatedly compares per-sample gradient vectors so that the subset's aggregate gradient approximates the full-dataset gradient, typically through a greedy orthogonal matching pursuit. Such matching is computationally expensive, sensitive to gradient noise, and scales with the dataset. GRAFT does not match gradients at all. It instead poses a purely geometric question, which handful of examples spans the largest volume of the batch's dominant feature subspace, and answers it in closed form with MaxVol. This volume criterion is provably near-optimal (within the Goreinov-Tyrtyshnikov factor) and coincides with the classical D-optimal experimental design, the statistically most informative subset for a linear model on those features, so GRAFT selects by a principled optimality criterion rather than by imitating a target gradient. Because selection operates on cheap low-rank features and never forms or compares gradients, its cost is independent of the dataset size, and gradient information re-enters only through the lightweight check that right-sizes the budget. In short, GRAFT replaces costly gradient approximation with a single near-optimal volume maximization, attaining the efficiency of one-shot pre-selection together with the adaptivity of integrated methods.

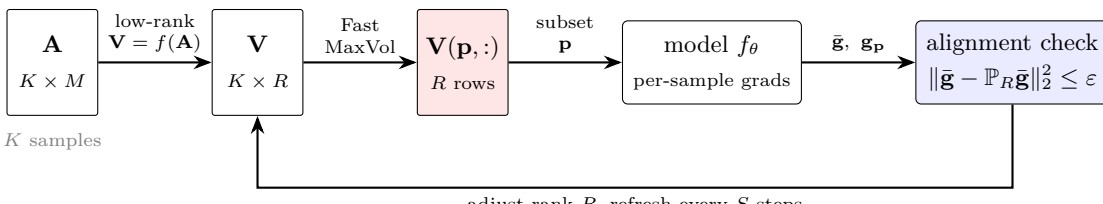

Figure 1: Overview of the GRAFT pipeline. Each batch $\mathbf{A} \in \mathbb{R}^{K \times M}$ is mapped by a low-rank feature extractor to $\mathbf{V} = f(\mathbf{A}) \in \mathbb{R}^{K \times R}$. Fast MaxVol then selects $R$ representative rows $\mathbf{p}$, and the subset gradient $\mathbf{g_p}$ is compared with the full-batch gradient $\bar{\mathbf{g}}$. When the projection error $\|\bar{\mathbf{g}} - \mathbb{P}_R \bar{\mathbf{g}}\|_2^2$ exceeds the tolerance $\varepsilon$ the rank $R$ is increased, and when alignment is strong it is reduced. Selection is refreshed every $S$ steps, and the chosen indices $\mathbf{p} = \{i_1, \ldots, i_R\}$ drive the parameter updates.

By refreshing the subset as training proceeds, GRAFT adapts to the evolving optimization landscape, and because the volume-maximizing criterion always selects examples that span the batch's dominant directions, the chosen subset stays representative even as its composition changes. Our experiments demonstrate strong performance across multiple datasets and architectures, reducing both training time and energy usage with minimal accuracy degradation. These findings position GRAFT as a practical tool for sustainable and efficient deep learning at scale.

**Contributions.**

- We cast in-training subset selection as low-rank row sampling and adapt Fast MaxVol to select the examples that span the dominant feature subspace, at a per-iteration cost of $\mathcal{O}(KR^2 + |\mathbf{R}| \, Rd)$ (the low-rank basis refresh, $\mathcal{O}(KdR)$, is amortized over the selection interval) that is independent of the dataset size $n$ because it operates within the optimizer's mini-batches (Section 3).

- We introduce a gradient-projection criterion that sets the subset size $R$ adaptively, enlarging it only when the selected gradients fail to span the full-batch gradient within a tolerance $\varepsilon$. As design motivation, a convergence analysis shows that controlling this projection error bounds the stationarity gap by $O(1/\sqrt{T}) + O(\varepsilon^2)$; we present this as justification for the criterion rather than as a tight or self-contained theoretical contribution (Section 3.3).

- We evaluate on six classification benchmarks up to ImageNet-1K, reporting accuracy alongside wall-clock, energy, and $CO_2$, and we explicitly characterize the regimes where GRAFT helps (aggressive pruning, large redundant batches) and where its overhead is not repaid (near-full fractions, severe class imbalance).

- We show the criterion is not vision-specific. It carries over to BERT fine-tuning, and, via a D-optimal-design reading of MaxVol, we give a *preliminary* demonstration that it also curates instruction-tuning data without labels, gradients, or GPU, selecting strictly more diverse, less redundant subsets than K-Means or random sampling (Section 4.2); a downstream fine-tuning study of those subsets is left to future work.

## 2  Related Works

Subset selection splits into pre-selection and integrated approaches. *Pre-selection* scores samples before training: EL2N and GraNd Paul et al. (2021) use early gradient norms, Forget Score Toneva et al. (2018) counts misclassifications (sensitive to noise and class imbalance), SVP Coleman et al. (2020) scores with a lightweight proxy whose fidelity bounds its quality, and DRoP Vysogorets et al. (2025) casts selection as distributionally robust optimization, robust to noise but costly through adversarial evaluations. Being fixed before training, these cannot follow the evolving dynamics. *Integrated* methods adapt the subset during training: GradMatch Killamsetty et al. (2021b) minimizes the gradient discrepancy $\|\nabla_\theta \mathcal{L}(S) - \nabla_\theta \mathcal{L}(\mathcal{X})\|_2^2$, while GLISTER Killamsetty et al. (2021a) and CRAIG Mirzasoleiman et al. (2020) use submodular objectives; benchmarks Guo et al. (2022) confirm these follow the dynamics but recompute and compare gradients every step, scaling with dataset size, and in the CORDS implementation run out of memory at large batch sizes. GRAFT sits between the two families: it runs online and adapts like the integrated methods, but selects by a geometric volume-maximisation criterion that never forms or compares gradients, keeping per-step cost close to the scoring family while staying responsive to the current loss landscape, and it adjusts subset size dynamically. Table 4 contrasts these methods along scalability and memory axes. Other related work includes BADGE Ash et al. (2020) (uncertainty with gradient-embedding clustering), moderately hard core-sets Xia et al. (2023), dataset distillation Yu et al. (2023) (which struggles at moderate sizes), the hybrid SelMatch Lee et al. (2024) (real-data selection refined by gradient matching, with extra meta-optimization), and SubSelNet Jain et al. (2023) for NAS (limited out-of-distribution). Most closely related in spirit to GRAFT's gradient-subspace view is LESS Xia et al. (2024), which selects instruction-tuning data for large language models by matching per-sample gradient features (computed via LoRA) to a target validation set. Where LESS maximises *relevance* to a held-out task distribution via nearest-neighbour retrieval in gradient space, GRAFT maximises *diversity* via volume (MaxVol), operating online within the training loop without requiring a held-out target. The two strategies are complementary. GRAFT provides a principled theoretical foundation (D-optimal selection under a frozen backbone, detailed in §4.2) for the gradient-aware data-selection family that LESS instantiates for the LLM fine-tuning regime.

**Preliminaries**  Let $\mathcal{X} = \{x_i\}_{i=1}^n \subset \mathbb{R}^d$ be a training dataset partitioned into $B$ mini-batches $\{\mathcal{X}_i\}_{i=1}^B$ of size $K \ll n$. For a batch $\mathcal{X}_i$, the gradient matrix $G \in \mathbb{R}^{d \times K}$ is defined as $G = [\nabla_\Theta L(\Theta; x_1), \ldots, \nabla_\Theta L(\Theta; x_K)]$. We project $\mathcal{X}_i$ into a low-rank embedding $V = f(\mathcal{X}_i) \in \mathbb{R}^{K \times R}$ ($R \ll d$) and use Fast Maxvol to select $R$ rows $\mathbf{p} = [p_1, \ldots, p_R]$ maximizing submatrix volume. The subset $G_R = G(:, \mathbf{p})$ approximates the full-batch gradient $\bar{g}$ via $\tilde{g} = \frac{1}{R} \sum_{r=1}^R G(:, p_r)$, with $\|\bar{g} - \tilde{g}\|_2 \leq \varepsilon$.

## 3  Data Point selection via Row Sampling

We select training points by row subset selection on the batch matrix $\mathbf{X} \in \mathbb{R}^{n \times d}$, in the spirit of Nyström and cross/skeleton approximation Mathur et al. (2023), which seek a submatrix that preserves the dominant structure rather than a global factorization. The classical Cross-2D method Tyrtyshnikov (2000) applies Maxvol Goreinov et al. (2010) to select rows *and* columns alternately by maximal submatrix volume, but the joint selection is costly and sensitive to initialization, and in our experiments it underperforms (Figure 5). We therefore reduce the problem to row sampling on an ordered feature matrix, in two stages: feature extraction then sample selection.

### 3.1 Sampling of Ordered Extracted Features

**Step 1. Feature Extraction.** For a batch $A \in \mathbb{R}^{K \times M}$ of $K$ samples, a feature map $f : \mathbb{R}^{K \times M} \to \mathbb{R}^{K \times R}$ ($R \ll M$) produces $V = f(A)$ whose rows $\mathbf{v}_k = V_{k,:}$ are per-sample features and whose columns are ordered by descending relevance, $\mathrm{Rel}(1) \geq \cdots \geq \mathrm{Rel}(R)$ (variance, mutual information with the labels, or correlation). We instantiate $f$ by SVD, $A = U\Sigma V^\top$, keeping the top-$R$ left singular vectors $U_R \in \mathbb{R}^{K \times R}$; $f$ may equally be a nonlinear encoder (e.g. a shallow autoencoder). The goal is a low-dimensional $V$ that captures the dominant structure for efficient selection.

**Step 2. Sample Selection using Fast MaxVol Method.** Given the feature matrix $\mathbf{V} \in \mathbb{R}^{K \times R}$, the goal is to sequentially select sample indices $p_r$ from the top-$R$ features such that the submatrix formed by the selected samples maximizes the volume, capturing the most significant information. One can apply the "conventional" Maxvol algorithm Goreinov et al. (2010), which iterates the row index selection until the entries of the interpolation matrix are below a predefined threshold (often close to 1). Alternatively, we show a fast Max-volume algorithm with $R$ iterations to select $R$ row indices. Let $\mathbf{p} = [p_1, \ldots, p_R]$ denote the set of selected row indices.

- **Select the first data point index.** Start with the first column $\mathbf{v}_1 = \mathbf{V}(:, 1)$ and select the index $p_1$ corresponding to the element with the maximum absolute value, i.e., the most dominant entry in the most relevant feature $p_1 = \arg\max_i |\mathbf{v}_1(i)|$

- **Selecting subsequent data point indices.** The second data point index, $p_2$, is selected such that the submatrix $\mathbf{V}([p_1, p_2], [1, 2])$ has the maximum volume, i.e., maximum absolute value of the determinant $p_2 = \arg\max_i |\det(\mathbf{V}([p_1, i], [1, 2]))|$ where the determinant of the submatrix $\mathbf{V}([p_1, i], [1, 2])$ is computed as

$$
\det(\mathbf{V}([p_1, i], [1, 2])) = \mathbf{v}_{p_1, 1} \mathbf{v}_{i, 2} - \mathbf{v}_{p_1, 2} \mathbf{v}_{i, 1}
$$
$$
= \mathbf{v}_{p_1, 1} \mathbf{r}_{i, 1} \tag{1}
$$

Here the residual $\mathbf{r}_1 = \mathbf{v}_2 - \mathbf{v}_1 (\mathbf{v}_{p_1, 1})^{-1} \mathbf{v}_{p_1, 2}$. The above projection nullifies the $p_1$ element in $\mathbf{r}_1$, i.e., $\mathbf{r}_1(p_1) = 0$. From (1), finding the index $i$, which maximizes the volume of $\mathbf{V}([p_1, i], [1, 2])$ in (3.1) is equivalent to identifying the maximum absolute element of $\mathbf{r}_1$, i.e., $p_2 = \arg\max_i |\mathbf{r}_1(i)|$. We next update the selected index set $\mathbf{p} = [p_1, p_2]$, and proceed with finding the other indices.

Assume the selected index set $\mathbf{p} = [p_1, p_2, \ldots, p_{j-1}]$, we need to select the $j$-th column index such that the submatrix $\mathbf{V}([\mathbf{p}, p_j], [1 : j])$ of $j$ rows and the first $j$ columns has the maximum volume, i.e.,

$$
p_j = \arg\max_i \left| \det \left( \mathbf{V}([\mathbf{p}, i], [1 : j]) \right) \right|.
$$

Suppose that $\mathbf{V}(\mathbf{p}, [1 : j - 1])$ is invertible, we define the residual term $\mathbf{r}_j = \mathbf{v}_j - \mathbf{V}(:, [1 : j - 1]) (\mathbf{V}(\mathbf{p}, [1 : j - 1]))^{-1} \mathbf{v}_{\mathbf{p}, j}$. By applying Sylvester's determinant theorem to the block matrix

$$
\mathbf{V}([\mathbf{p}, i], [1 : j]) = \begin{bmatrix} \mathbf{V}(\mathbf{p}, [1 : j - 1]) & \mathbf{v}_{\mathbf{p}, j} \\ \mathbf{v}_{i, 1:j-1} & \mathbf{v}_{i, j} \end{bmatrix},
$$

the determinant simplifies to $\det \left( \mathbf{V}([\mathbf{p}, i], [1 : j]) \right) = \det \left( \mathbf{V}(\mathbf{p}, [1 : j - 1]) \right) \cdot \mathbf{r}_j(i)$. Hence, finding the most relevant index $p_j$ reduces to selecting the index corresponding to the maximum absolute value of $\mathbf{r}_j$ $p_j = \arg\max_i |\mathbf{r}_j(i)|$.

*Remark* 1 (MaxVol selects rows spanning the dominant subspace). Let $A \in \mathbb{R}^{K \times M}$ be a batch matrix with SVD $A = U\Sigma V^\top$, let $V = U_R \in \mathbb{R}^{K \times R}$ be its top-$R$ left singular vectors, and let $S \subset [K]$, $|S| = R$, be the rows returned by MaxVol applied to $V$. MaxVol picks the rows whose submatrix $V(S, :)$ has near-maximal volume, so the selected rows span the dominant rank-$R$ subspace of $A$, and the discarded component $A - A_R$ has spectral norm $\sigma_{R+1}$. Consequently, when the batch is approximately low-rank ($\sigma_{R+1}$ small relative to $\sigma_1$), the selected rows capture the directions carrying most of the batch's energy, and with them most of the variation in the per-sample gradients. We deliberately stop short of a closed-form bound on the full-batch-versus-subset gradient gap. For a *nonlinear* gradient map $g(x) = \nabla_\theta L(\theta; x)$ the unweighted subset

---

**Algorithm 1:** Training with Gradient-Aligned Sampling

---

**Input:** Training dataset $\mathcal{X}$, ranks $\mathbf{R} = \{R_i\}_{i=1}^c$, feature matrices $\mathbf{V}$, selection period $S$, and batch size $K$
**Output:** Sampled data subset indices $\mathcal{S}^t$
**for** $t = 1, \ldots, T$ **do**
    **Stage 1. Subset Selection**
    **if** $t \bmod S == 0$ **then**
        **for** *each batch* $\mathcal{X}_i \subset \mathcal{X}$ **do**
            $\bar{\mathbf{g}}_i \leftarrow \frac{1}{K} \sum_{k=1}^K \nabla_\Theta L(\Theta^t; \mathcal{X}_i(:, k))$
            **for** $R_i \in \mathbf{R}$ **do**
                $\mathcal{S}_i^r = \text{fast-maxvol}(\mathbf{V}_i, R_i)$
                $\mathbf{G}_{R_i} = [\ldots, \nabla_\Theta L(\Theta^t; \mathcal{X}_i(:, \mathcal{S}_i^r(p))), \ldots]$
                $d_{R_i} = \|\bar{\mathbf{g}}_i - \mathbf{G}_{R_i} \mathbf{G}_{R_i}^\dagger \bar{\mathbf{g}}_i\|_2^2$
            $R^* = \arg\min_{R_i} \{d_{R_i}\}_{i=1}^{|\mathbf{R}|}$
            $\mathcal{S}_i = \mathcal{S}_i^{R^*}$
        $\mathcal{S}^t = \{\mathcal{S}_i\}_{i=1}^I$
    **else**
        $\mathcal{S}^t = \mathcal{S}^{t-1}$
    **Stage 2. Model Update**
    Update all model parameters using $\mathcal{S}^t$

---

mean need not equal the full-batch mean even when $\sigma_{R+1} = 0$ (an exactly rank-$R$ batch whose rows occupy distinct points of the subspace already gives $\bar{g} \neq g_S$), so no $\sigma_{R+1}$-only bound on that gap can hold. The quantity that actually controls gradient fidelity is the projection residual of Lemma 1, which GRAFT keeps small by choosing *which* rows enter the subspace and which we measure directly (Fig. 2).

### 3.2 Dynamic Gradient based refinement

A key challenge in the proposed algorithm is determining the optimal number of representative samples to select. Because Fast MaxVol applied to a rank-$R$ feature basis returns exactly $R$ rows, the working rank and the selected subset size are the same quantity throughout the paper; we therefore denote both by $R$, which is an intentional identification rather than an overloaded symbol. This number must balance the size of the subset against the potential loss in accuracy. We propose selecting $R$ such that the gradient direction computed from the selected samples closely aligns with the gradient direction of the per-iteration full batch gradients. The rank $R$ can either be fixed or dynamically updated every $S$ training iterations, where $S$ is typically set to values between 20 to 50. In fixed-rank/fixed sample-size schemes, the subset size $R$ is predetermined using heuristics or a pre-defined metric such as in Killamsetty et al. (2021b) and remains constant throughout training. However, this approach fails to consider the evolving nature of the optimization landscape. GRAFT dynamically selects $R$ from a set of candidates to minimize the projection error $\|\bar{g} - P_R \bar{g}\|_2^2$, selecting $R$ that satisfies a predefined error threshold $\epsilon$ Bottou et al. (2018).

**Gradient Approximation with Sampled Subsets** The training dataset $X$ is randomly partitioned into batches $X_i$, each of size $m \times K$, where $K$ is the number of data points in the batch. The batch gradient $\bar{g}$ is calculated as the average gradient vector for the batch

$$\bar{g} = \nabla_\Theta \mathcal{L}(\Theta; X_i) = \frac{1}{K} \sum_{k=1}^K \nabla_\Theta \mathcal{L}(\Theta; X_i(:, k)).$$

This gradient serves as the reference direction. Since gradients from fewer samples may result in less precise approximations, the goal is to select a minimal number of dominant samples $R$ such that their gradients, denoted $G_R = [g_1, g_2, \ldots, g_R]$, approximate the batch gradient. Here, $g_r = \nabla_\Theta \mathcal{L}(\Theta; X_i(:, p_r))$ for $r = 1, \ldots, R$.

The key requirement is that $\bar{g}$ lies in the subspace spanned by $G_R$ $R = \arg\min_R d(\bar{g}, G_R)$, where $d(\bar{g}, G_R)$ can be defined as angular error $d(\bar{g}, G_R) = \arcsin(\|\tilde{g} - \tilde{G}_R \tilde{G}_R^T \tilde{g}\|_2) = \arcsin\left(\sqrt{1 - \|\tilde{G}_R^T \tilde{g}\|_2^2}\right)$, or projection error $d(\bar{g}, G_R) = \|\tilde{g} - \tilde{G}_R \tilde{G}_R^T \tilde{g}\|_2^2 = 1 - \|\tilde{G}_R^T \tilde{g}\|_2^2$, Here, $\tilde{g}$ is the unit vector of the reference gradient $\bar{g}$, and $\tilde{G}_R$ is an orthogonal basis of the gradient matrix $G_R$ Bottou et al. (2018); Halko et al. (2011).

**Dynamic Rank Adjustment** We propose dynamically selecting $R$ to control projection error $\|\bar{g} - P_R \bar{g}\|_2^2$. Specifically, we search over candidate ranks $R_1 < \cdots < R_m$ and choose $R^* = \arg\min_{R_i} \|\bar{g} - P_{R_i}\bar{g}\|_2^2$, subject to $\|\bar{g} - P_{R_i}\bar{g}\|_2 \leq \epsilon$, on the current batch during the selection step $S$. By Corollary 1, bounding the projection error ensures convergence under standard smoothness and variance assumptions Bottou et al. (2018).

We state the following results to *motivate* GRAFT's design, instantiating standard inexact-gradient arguments for the case where the inexactness is the MaxVol projection error. They are not a tight or novel analysis, and none of GRAFT's empirical claims depend on their constants; the operative quantity is the measured projection residual of §4. Theorem 2 concerns the idealised projected direction $\text{Proj}_{\mathcal{S}_R}(\bar{g})$, which GRAFT realises approximately through the unweighted subset-mean gradient; we track the deviation through the same measured residual rather than bound it in closed form.

GRAFT's convergence rests on one inexact-gradient guarantee. If the MaxVol-selected subspace $\mathcal{S}_R$ keeps the projection error $\|\bar{g} - \text{Proj}_{\mathcal{S}_R}(\bar{g})\| \leq \epsilon$, then descent along the projected direction reaches a stationary neighborhood whose radius is controlled by $\epsilon$ (Theorem 2). This error has the closed form $\|\bar{g} - \tilde{G}_R \check{G}_R^\top \bar{g}\|_2^2 = \|\bar{g}\|_2^2 (1 - \|\tilde{G}_R^\top \bar{g}\|_2^2 / \|\bar{g}\|_2^2)$ (Lemma 1), so the dynamic-rank rule, which adjusts $R$ to hold this error below $\epsilon$, preserves the same $\epsilon$-controlled convergence (Corollary 1). Statements and proofs are deferred to Appendix 9.

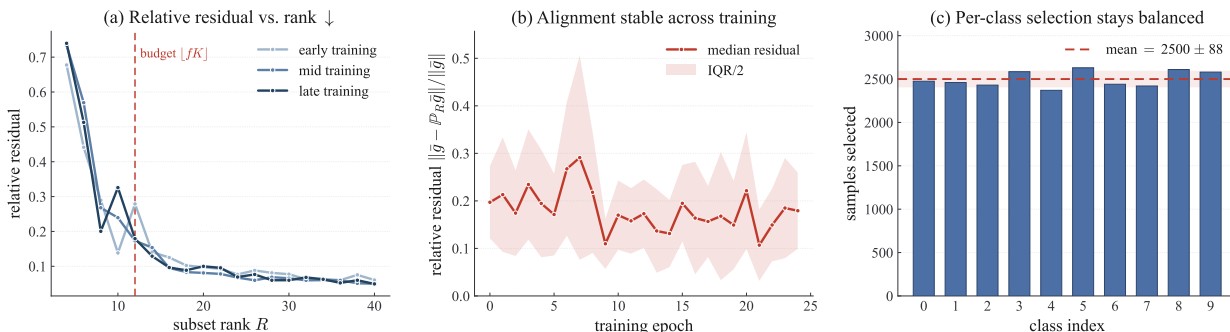

Figure 2: Selection dynamics during training, measured on the real GRAFT pipeline (ResNet-18, CIFAR-10, $f$=0.10, batch $K$=128; selection refreshed every five epochs, probed each epoch). **(a)** GRAFT's selection criterion, the relative projection residual $\|\bar{g} - \mathbb{P}_R \bar{g}\|/\|\bar{g}\|$ versus subset rank $R$ at three training stages; it drops steeply and saturates past an elbow near the operating budget $\lfloor fK \rfloor$ (red line), so a $\sim 9\%$ subset already captures most of the full-batch gradient direction. **(b)** At the budget the relative residual stays low and roughly constant ($\approx 0.15$ to $0.25$) across training, i.e. GRAFT *maintains* the $\epsilon$-approximation of Cor. 1 rather than degrading. **(c)** The number of samples selected per class stays balanced across training.

**Gradient Alignment and Rank Evaluation** Figure 2(a,b) logs our reference GRAFT run (ResNet-18/CIFAR-10, $f$=0.10) through the scale-free relative residual $\|\bar{g} - \mathbb{P}_R \bar{g}\|/\|\bar{g}\| \in [0, 1]$ bounded in Lemma 1; normalising by $\|\bar{g}\|$ is needed because the raw residual shrinks with the falling gradient magnitude rather than from improved alignment. Against rank $R$ (panel a) the residual drops steeply and saturates past an elbow near $R \approx 8$ to $12$, so the operating budget $\lfloor fK \rfloor \approx 12$ already lies in the flat region and a $\sim 9\%$ subset captures most of the full-batch gradient direction (residual $\approx 0.18$). The curves at three training stages nearly coincide, and at the budget the residual stays low and roughly constant ($\approx 0.15$ to $0.25$; panel b), so GRAFT *maintains* the $\epsilon$-approximation of Corollary 1 by *which* samples it selects rather than by spending more of them. Selection also stays class-balanced throughout (panel c; 2370 to 2630 per class, mean 2500), so no class is starved under aggressive pruning.

### 3.3 Complexity Analysis

We use three standard assumptions: **(A1)** the loss $\mathcal{L}$ is $L$-smooth; **(A2)** the stochastic gradients $\bar{g}_i$ are unbiased with bounded variance; and **(A3)** the projection error satisfies $\|\bar{g}_i - G_R G_R^\dagger \bar{g}_i\|_2 \leq \varepsilon$ at every step, which is precisely the condition GRAFT enforces through its dynamic-rank rule.

*Theorem* 1 (Projected-gradient convergence). Under (A1) to (A3), SGD with projected directions $G_R G_R^\dagger \bar{g}_i$ satisfies

$$\min_{t \leq T} \mathbb{E}\big\|\nabla L(\Theta_t)\big\|_2^2 \ \leq \ O\Big(\tfrac{1}{\sqrt{T}}\Big) \ + \ O(\varepsilon^2).$$

*Proof sketch.* With $L$ smooth and stochastic gradients unbiased with bounded variance, the standard SGD descent recursion yields an $O(1/\sqrt{T})$ rate. The projection step introduces a deterministic bias whose energy is controlled by the bound $\|\bar{g}_i - G_R G_R^\dagger \bar{g}_i\|_2^2 \leq \varepsilon^2$. This adds an $O(\varepsilon^2)$ term to the stationarity gap while leaving the stochastic term unchanged; see the Supplement (Detailed Proofs) for a full derivation. $\qquad\square$

**Computational complexity** Per iteration, Fast MaxVol selection with gradient-projection evaluation runs in

$$O\big(KR^2\big) \ + \ O(|\mathbf{R}|\,R\,d)$$

time and uses $O(Kd + dR + R^2)$ memory. Computing gradients for the selected $R$ samples contributes $O(Rd)$ and is included when reported separately. A basis/feature refresh (performed periodically) costs $O(KdR) + O((K+d)R^2)$ when it occurs and is amortized over the refresh interval. Because all operations act on the current mini-batch, no term scales with the dataset size $N$; the full derivation is in the Supplement (Complexity Analysis). $\qquad\square$

In the worst case the dynamic-rank rule selects the largest candidate rank $R_{\max}$ at every refresh; substituting $R \to R_{\max}$ ($R_{\max} \leq 64$ in our runs) and amortizing the basis rebuild over the $S$ reuse steps ($S$ between 40 and 50) leaves a per-iteration overhead of $O\big(\tfrac{1}{S}(KdR_{\max} + |\mathbf{R}|R_{\max}d)\big)$, still independent of $N$ and a small fraction of one forward-backward pass. Because the rule never exceeds $R_{\max}$, a poor selection at most drives the subset toward the full batch, so GRAFT degrades to slightly above full-batch cost rather than diverging, and any mis-selection persists for at most $S$ steps before the next alignment check corrects it. Selection therefore helps most when batches are redundant and approximately low-rank, and we recommend GRAFT for data fractions below about 35% with batch sizes of at least 128, where a small subset already spans most of the full-batch gradient direction (Fig. 2(a,b)); full-data or one-shot pre-selection is preferable outside this regime.

## 4 Experiments

GRAFT is evaluated in three regimes. We measure efficiency on six image-classification benchmarks (§4.1), transfer to language-model fine-tuning, with a preliminary extension to training-free instruction-data curation (§4.2), and ablate the feature extractor and the selector (§4.3).

### 4.1 Efficiency on vision benchmarks

We report raw Top-1 accuracy and measured $CO_2$ emissions for each method across data fractions, and we visualize the accuracy and emissions trade-off in Fig. 3. Full per-dataset tables are in the supplement. We compare GRAFT against full-data training and three integrated subset-selection baselines (GLISTER Killamsetty et al. (2021a), GradMatch Killamsetty et al. (2021b), and DRoP Vysogorets et al. (2025)), and we add random sampling and the GradMatch per-batch family at ImageNet-1K scale, where GLISTER and DRoP do not fit in memory on our hardware. $CO_2$ is estimated with the eco2AI library Budennyy et al. (2022) as $\mathcal{E} = P \times t \times I$, where $P$ is hardware power (kW), $t$ is training time (h), and $I$ is the grid carbon intensity (kgCO$_2$/kWh).

**Cold- vs. warm-start and fairness of comparison.** We report two variants. GRAFT (cold) trains from random initialization and is the strict apples-to-apples counterpart to the baselines, all of which are likewise cold-started; GRAFT WARM additionally initializes the feature extractor from a short full-data

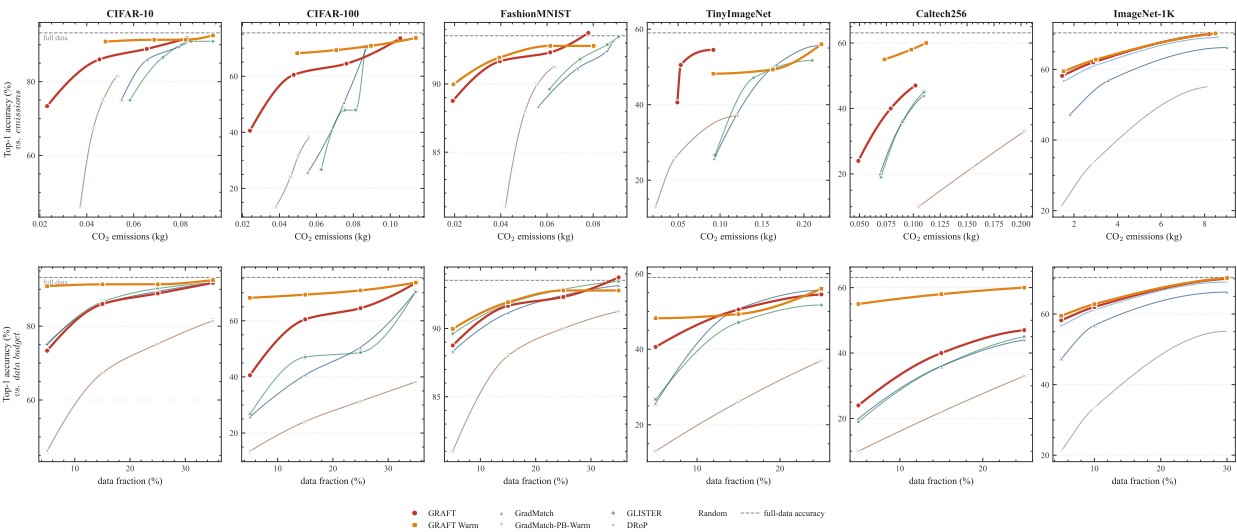

Figure 3: Accuracy and efficiency trade-offs across six benchmarks. **Top row.** Top-1 accuracy against $CO_2$ emissions (linear axes), an accuracy versus cost Pareto view; all markers are measured operating points, and the connecting curves are a monotone (non-decreasing) interpolation through them, since accuracy can only rise with more budget. GRAFT (red) and GRAFT WARM (orange) are drawn as connected operating curves and trace the upper-left Pareto frontier; the gradient-matching baselines (lighter markers) lie below or to the right and are dominated. Random sampling adds no selection cost and is therefore the cheapest method at a fixed fraction, so it is not dominated on the easiest benchmarks; GRAFT matches it there and overtakes it on accuracy under aggressive pruning. **Bottom row.** Top-1 accuracy against data fraction (5% to 35%, or 5% to 25% for TinyImageNet and Caltech256, and 5% to 30% for ImageNet-1K), with the same monotone (non-decreasing) interpolation through the measured points for every method. The dashed grey line marks full-data accuracy (both rows). GRAFT's margin is clearest under aggressive pruning. At ImageNet-1K scale we compare against the methods that fit in memory on our hardware (random sampling and the GradMatch per-batch family).

warm-up and is presented as an optional enhancement rather than as the basis of our main efficiency claims. Accordingly, comparisons against GLISTER, GradMatch, and DRoP should be read primarily against cold GRAFT, while GRAFT WARM quantifies the additional accuracy attainable when a brief full-data pass is affordable.

**Results.** We report the head-to-head numbers directly; full per-dataset tables (accuracy and $CO_2$ at $f \in \{0.05, 0.15, 0.25, 0.35\}$) are in the supplement (Tables 6 to 10), and Fig. 3 visualizes the resulting frontiers. The consistent finding is that at matched accuracy GRAFT sits at the low-emission end of the frontier, with its largest margins under aggressive pruning. On CIFAR-100 at $f = 0.05$, cold GRAFT reaches 40.6% versus 25.7% (GradMatch) and 26.7% (GLISTER) at comparable or higher emissions; on TinyImageNet at $f = 0.05$ it reaches 40.6% versus $\leq 26.7\%$ for every baseline while emitting the least $CO_2$ (0.049 kg). On ImageNet-1K it comes within 0.36 points of full-data ResNet-18 accuracy (70.0% vs. 70.36%) at $f = 0.30$ for 8.20 kg rather than 27.63 kg (Table 10). We are deliberately precise about where the win is *not* an accuracy win. On the easier CIFAR-10 benchmark the gap to the strongest baselines is small and not always in GRAFT's favor (e.g. at $f = 0.25$ cold GRAFT trails GLISTER, 88.9% vs. 90.2%), and GRAFT's contribution there is reduced emissions at equal accuracy *relative to the gradient-matching baselines*, not relative to random sampling, which is cheaper still. A multi-seed analysis makes this precise. On CIFAR-10, per-batch GRAFT and random sampling are statistically indistinguishable in mean accuracy at every fraction (Welch $p \geq 0.47$; Appendix 10), so on this easy benchmark random is the more economical choice and GRAFT's advantage is confined to undercutting the costlier integrated methods. The warm-start variant adds the most at very low emissions ($\mathcal{E} < 0.05$ kg), where initializing the extractor from a brief full-data pass lifts low-fraction accuracy substantially (CIFAR-10 $f = 0.05$: 90.9% vs. 73.4% cold).

Table 1: $CO_2$ emissions (kg) and accuracy (%) for BERT fine-tuned on IMDB.

| Method | Emiss. (kg) | Top-1 Acc. (%) |
|---|---|---|
| Full (Baseline) | 0.32 | 93.92 |
| GRAFT (10%) | 0.05 | 91.72 |
| GRAFT Warm (10%) | 0.14 | 93.74 |
| GRAFT (35%) | 0.15 | 93.56 |
| GRAFT Warm (35%) | 0.19 | 93.71 |

**Cross-dataset summary.** The method ranking is consistent across all six benchmarks (Fig. 3): GRAFT and GRAFT WARM occupy the low-emission frontier and close most of the gap to full-data accuracy by $f = 0.35$. Relative to the nearest *gradient-matching* baseline at matched accuracy, GRAFT cuts $CO_2$ by 0.15 kg on CIFAR-10 Krizhevsky & Hinton (2009), 0.21 kg on TinyImageNet Fei-Fei et al. (2015), and 0.28 kg on Caltech256 Griffin et al. (2007), and it leads every such baseline at each fraction up to ImageNet-1K scale (Table 10). Random sampling is cheaper than GRAFT at any fixed fraction, since it adds no selection cost; GRAFT's advantage over random is accuracy, decisive under aggressive pruning and within noise on the easiest benchmarks.

### 4.2 Transfer to language-model fine-tuning and data curation

The integrated baselines above (GLISTER, GradMatch, DRoP) were built for, and have almost exclusively been demonstrated on, supervised image classification. GRAFT's criterion is modality-agnostic, acting on feature embeddings, so it carries over to text unchanged. We show this in two settings, fine-tuning a transformer and curating an instruction-tuning pool without any training, the regime that the modern data-centric LLM literature Xia et al. (2024) targets but that the classical coreset baselines do not address.

**Fine-tuning BERT on IMDB.** Table 1 reports BERT fine-tuned on IMDB. Cold GRAFT on 35% of the data reaches 93.56% accuracy at 0.15 kg $CO_2$, within 0.36 points of full-data training (93.92% at 0.32 kg) for under half the emissions; GRAFT WARM narrows the gap further to 93.71% at 0.19 kg. At an aggressive 10% budget, warm-starting matters most, lifting accuracy from 91.72% (cold) to 93.74% by reusing pretrained representations, at 0.14 kg against 0.05 kg. The efficiency-frontier behaviour observed on vision thus carries over to transformer fine-tuning.

**Training-free instruction-data curation (a preliminary extension).** We close with a smaller, preliminary study that probes how far GRAFT's selection criterion reaches beyond gradient features and beyond vision. It is a transfer demonstration rather than a full instruction-tuning evaluation, and should be read as such. GRAFT's MaxVol criterion admits a clean statistical interpretation that motivates applying it directly to instruction embeddings. When the backbone is frozen and only a linear head $W$ is updated, the per-sample feature vectors $\phi(x_i)$ are fixed and head-fitting reduces to a standard linear regression $\min_W \|\Phi W - Y\|_F^2$. The D-optimal experimental design criterion Kiefer & Wolfowitz (1960) selects the size-$R$ subset $S^*$ that maximises $\det(\Phi_S^\top \Phi_S)$, minimising the volume of the confidence ellipsoid for $\hat{W}$. Since Fast MaxVol maximises exactly this determinant (up to the Goreinov-Tyrtyshnikov approximation ratio $e^n$), *GRAFT is the D-optimal data selector for any linear adapter on a frozen backbone*, including LoRA layers applied to the final projection, prefix-tuning vectors, and BitFit bias terms.

The D-optimal connection above predicts that GRAFT's MaxVol criterion should select maximally diverse examples even when applied *directly to sentence embeddings* rather than gradient features, a fully supervised-free and GPU-free regime applicable to LLM instruction data curation. To test this, we embed $N$=8,000 instructions from Alpaca-52k Taori et al. (2023) using `sentence-transformers/all-MiniLM-L6-v2` ($d$=384, unit-norm). We apply *global* MaxVol, a single compact SVD of $\Phi^\top \in \mathbb{R}^{d \times N}$ followed by Fast MaxVol on the right-singular-vector matrix, selecting the $R \leq d$ columns whose submatrix has maximum volume. This is the provably D-optimal subset for the fixed-embedding linear-regression problem. We compare against Mini-Batch K-Means (one representative per centroid) and random sampling at fractions $f \in \{0.5\%, 1\%, 2\%, 3\%, 4\%\}$, the natural regime where $R \leq d$=384.

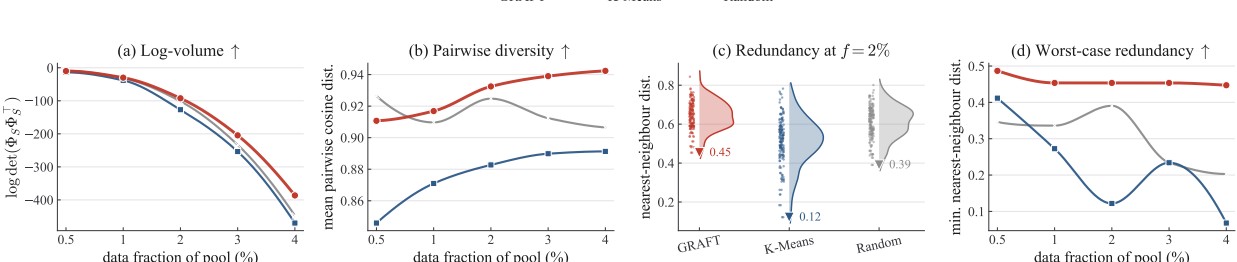

Figure 4: Instruction-space diversity on 8,000 Alpaca-52k instructions embedded with `all-MiniLM-L6-v2`, comparing global MaxVol (GRAFT criterion), Mini-Batch K-Means, and random sampling. **(a)** Log-volume $\log \det(\Phi_S \Phi_S^\top)$, the quantity MaxVol maximises, and **(b)** mean pairwise cosine distance, an independent diversity measure, both versus data fraction; GRAFT leads at every fraction at and above 1% with a margin that widens as the budget grows (at 0.5% all three are within noise, as 40 near-unit vectors in $d{=}384$ dimensions are nearly orthogonal under any rule). **(c)** Distribution of each selected example's cosine distance to its nearest other selected example at $f{=}2\%$ ($R{=}160$); the marker is each method's minimum, its closest, most redundant pair. **(d)** That minimum versus budget. GRAFT holds a high floor ($\approx 0.45$) at every fraction, so it never keeps a near-duplicate pair, whereas the K-Means and random floors fall to 0.07 and 0.20 by 4%.

Figure 4 shows the results. For every budget at and above $f{=}1\%$, Global MaxVol achieves the highest *log-volume* $\log \det(\Phi_S \Phi_S^\top)$ (Fig. 4a), which is expected since it directly optimises this quantity, and the margin over both baselines widens monotonically with the fraction. Crucially, over the same range it also achieves the highest *mean pairwise cosine distance* (Fig. 4b), an independent diversity metric not optimised by the selection procedure: 0.917 vs. 0.910 (Random) vs. 0.871 (K-Means) at $f{=}1\%$, rising to 0.942 vs. 0.907 vs. 0.891 at $f{=}4\%$. At the most aggressive 0.5% budget the three methods are statistically indistinguishable (random is marginally ahead), because so few points in a 384-dimensional embedding space are near-orthogonal under any selection rule; GRAFT's advantage emerges precisely once the budget is large enough that unstructured sampling begins to draw redundant, near-duplicate instructions. Beyond these aggregate scores, GRAFT's selections are also the least *redundant* (Fig. 4c,d). At $f{=}2\%$ the closest pair of selected instructions is 0.45 apart in cosine distance, against 0.12 for K-Means and 0.39 for random, and this minimum stays near 0.45 at every budget while the K-Means and random floors fall to 0.07 and 0.20 by $f{=}4\%$. Maximising volume therefore guarantees a mutually distinct subset in which no two near-duplicate instructions are both kept, whereas the baselines increasingly retain near-duplicates as the budget grows, precisely the failure mode that wastes fine-tuning compute on redundant supervision. Notably, topic-cluster coverage alone does not separate the methods (all three cover the 15 clusters with comparable entropy $\approx 0.97$); the advantage is visible only at this finer, within-subset scale. The same volume criterion that GRAFT applies to gradient features inside the training loop thus also curates a static pool in a single pass, extending its reach from in-training pruning to data selection for language-model fine-tuning. We are explicit about the scope of this experiment. It measures the *selection quality* of the chosen subsets (diversity and non-redundancy in embedding space), not end-to-end fine-tuned accuracy. Establishing that these better-curated subsets translate into better fine-tuned models is the natural next step and is left to future work; we include the result here as evidence that GRAFT's geometric criterion transfers beyond vision, not as a complete instruction-tuning study.

### 4.3 Ablations and analysis

**Feature-extractor ablation.** We compare three choices for the extractor $f$, namely Singular Value Decomposition (SVD), a shallow autoencoder (AE), and Independent Component Analysis (ICA), in two complementary settings on CIFAR-10. The first setting is end-to-end training with a ResNet-18 backbone on 25% of the data (Figure 5a). Here SVD reaches 90.3% test accuracy, ahead of AE (86.7%) and ICA (80.8%), reflecting SVD's strength at preserving the dominant batch structure that drives the gradient. The second setting is a linear-probe microbenchmark that isolates extractor quality and cost by classifying the extracted

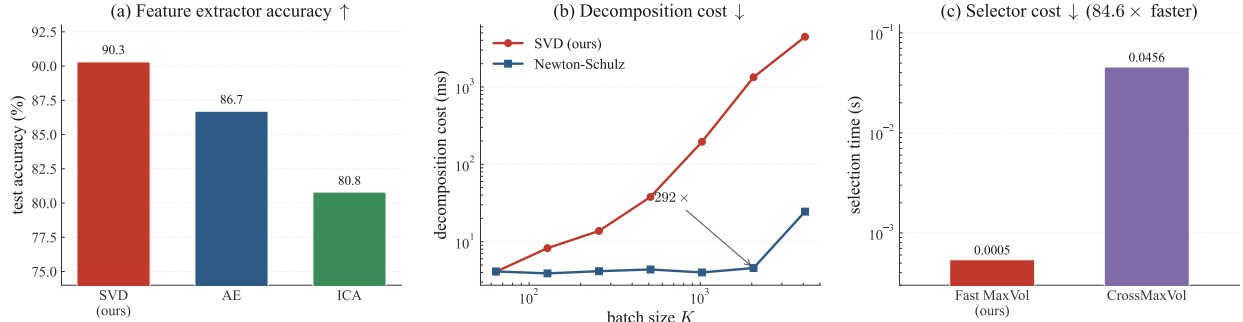

Figure 5: Ablation of GRAFT's two components, all values measured. **(a)** End-to-end test accuracy of the feature extractor (SVD, AE, ICA) with a ResNet-18 backbone on 25% of CIFAR-10; SVD is most accurate. **(b)** Decomposition cost versus batch size. The exact SVD scales cubically while a randomized Newton-Schulz basis stays flat (up to 292× cheaper); yet the Newton-Schulz basis yields near-random MaxVol selections (Jaccard ≈ 0.08 to the SVD selection), so SVD remains the default (see text and Table 2). **(c)** Selection time of Fast MaxVol against the CrossMaxVol baseline on Iris (which isolates the selection step); Fast MaxVol is 84.6× faster. Subspace-similarity numbers are reported in Table 2.

Table 2: Component-level ablations of GRAFT, all values measured. The *Quality* column is linear-probe top-1 accuracy for the feature extractor (five seeds, fixed 20% holdout, CIFAR-10) and subspace similarity for the selector (selected samples to the optimal subspace, Iris) and the basis (Newton-Schulz basis to the exact SVD basis, CIFAR-10, $K$=2048); *Time* is the corresponding per-batch cost. AE's probe gain over SVD is significant ($p$=0.007) while ICA's is not ($p$=0.65). Although the Newton-Schulz basis spans nearly the same subspace as the SVD (0.84), the MaxVol selection it induces matches SVD's only at chance (Jaccard ≈ 0.08), so the exact SVD remains GRAFT's default.

| Component | Method | Quality | Time (s) |
|---|---|---|---|
| Feature extractor | SVD ($R$=64) | $38.19 \pm 0.24\,\%$ | $0.0385 \pm 0.0025$ |
| | AE | $38.99 \pm 0.21\,\%$ | $0.1996 \pm 0.0105$ |
| | ICA | $38.25 \pm 0.00\,\%$ | $0.3921 \pm 0.0512$ |
| Selector (Iris) | Fast MaxVol (ours) | **0.6250** | **0.0005** |
| | CrossMaxVol | 0.5938 | 0.0456 |
| Basis (CIFAR-10) | Full SVD | 1.000 | 1.334 |
| | Newton-Schulz | 0.840 | **0.005** |

features with a lightweight logistic regression (five seeds 42 to 46, fixed 20% holdout, Table 2). The probe deliberately uses a weak classifier, so its absolute accuracies of around 38% lie far below the end-to-end numbers and serve only to rank the extractors at matched cost. AE is marginally most accurate (38.99%) but 5× slower (0.1996 against 0.0385 s/batch), and ICA is slowest (0.3921 s/batch) at 38.25%. Taken together, SVD wins the setting that matters, end-to-end accuracy, while also being the cheapest extractor, so we adopt it as GRAFT's default. AE's sub-one-point probe edge does not justify its 5× per-batch overhead inside the training loop.

**Subspace Similarity and Efficiency** The results presented in Table 2 compare the Subspace Similarity and Execution Time for the proposed Fast MaxVol and the baseline CrossMaxVol, implemented via the `teneva` AndreiChertkov (2025) library on the Iris dataset Fisher (1936), a controlled low-dimensional benchmark used here to isolate the selection step. Subspace similarity is quantified as the sum of squared cosines of the principal angles between the subspaces spanned by the selected samples $(V_1, V_2) = \sum_{i=1}^{k} \cos^2(\theta_i)$, where $V_1$ and $V_2$ are the subspace bases, and $\theta_i$ are the principal angles. Fast MaxVol achieves a similarity of 0.625, outperforming CrossMaxVol (0.59375), indicating that the subspace spanned by Fast MaxVol is more

aligned with the optimal representation. It is also far cheaper. Its selection step runs in 0.000539 seconds against 0.045594 seconds for CrossMaxVol (Figure 5c and Table 2), a speedup of approximately 84.6×.

**Is an exact SVD necessary?** GRAFT's selector consumes only an orthonormal basis of each batch's dominant rank-$R$ subspace, so in principle the exact SVD could be replaced by any cheaper basis. We test the most attractive alternative, a randomized sketch orthogonalized by the matmul-only cubic Newton-Schulz iteration. It is far cheaper, its cost staying essentially flat in the batch size $K$ ($\approx 4$ ms) while the SVD grows cubically (up to 292× faster at $K{=}2048$; Fig. 5b), and it recovers most of the dominant subspace (similarity $\approx 0.84$ to the SVD basis; Table 2). *Yet the selection collapses.* The MaxVol indices it produces overlap the full-SVD selection at a Jaccard of only $\approx 0.08$, essentially the random-chance level ($\approx 0.05$), and adding iterations does not help. Fast MaxVol selects by maximal-volume pivoting, which depends on the precise orientation of the basis vectors and not merely the subspace they span, so the $\sim 16\%$ residual subspace error Newton-Schulz leaves is enough to drive the pivots to a near-random subset. The exact SVD is therefore GRAFT's right default, not because it is fast but because Fast MaxVol needs the near-exact subspace that only an exact decomposition supplies; this is also why GRAFT's accuracy tracks feature-extractor quality (§5). The cost is affordable because selection runs on small per-batch matrices amortized over the refresh interval $S$.

## 5 Limitations and Future Work

GRAFT's performance depends on the quality of low-rank feature extraction, and the dynamic rank adjustment adds some overhead. Its batch-level subspace approximation is least robust under highly non-IID or imbalanced data, where small or skewed batches yield unstable selections, and accuracy degrades under extreme compression; at 5% of CIFAR-100 top-1 falls from roughly 75% to 41%, a clear limit to how aggressively data can be pruned. Our analysis is deliberately limited, characterizing convergence only to a stationary neighborhood whose radius scales with the tolerance $\varepsilon$ and giving no closed-form bound relating the subset gradient to the full-batch gradient (Remark 1 explains why a $\sigma_{R+1}$-only bound cannot hold for a nonlinear gradient map); a sharper analysis is left to future work. Finally, GRAFT WARM's gains presuppose an affordable full-data warm-up, so the cold-start variant remains the right comparison when no full-data pass is available.

**Beyond vision: PEFT and LLM agents.** The D-optimal interpretation in §4.2 bridges GRAFT to instruction-data selection for large language models Xia et al. (2024), and the embedding-space experiment above shows the criterion already transfers without labels or gradients. Extending it to the sequential, multi-layer PEFT regime, where $\phi(x_i)$ is aggregated across token positions and adapter gradients span several layers, is a natural next step. Because the volume criterion acts on any embedding space, it also applies to curating redundant self-generated agent trajectories and to volume-based context selection for retrieval-augmented agents, a principled alternative to similarity-based top-$k$ retrieval that returns near-duplicates. We leave an empirical study of these to future work.

## 6 Conclusion

We present GRAFT, a data sampling framework that selects training subsets by maximising the volume they span in a low-rank feature subspace, the criterion that coincides with D-optimal experimental design, and that adjusts the subset size online from a projection-residual check rather than by matching gradients. Integrated into the training loop at a selection cost independent of dataset size, GRAFT traces the accuracy-emissions Pareto frontier across six vision benchmarks, with its largest gains under aggressive pruning, and the same volume criterion carries over to BERT fine-tuning, with a preliminary extension to training-free curation of instruction-tuning data. We are deliberate about its limits. On easier benchmarks the cold-start variant only matches random sampling, which is cheaper because it adds no selection overhead, so GRAFT's emission savings there are measured against the costlier integrated baselines rather than against random, while its accuracy gain over random emerges only under aggressive pruning on harder datasets. GRAFT is a scalable step toward sustainable machine learning for resource-constrained training and AutoML pipelines, and we will release an open-source implementation.

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

## Supplementary Material

## 7   Notations

Table 3: Summary of notation used throughout the paper

| Symbol | Description | Dim./Type |
|---|---|---|
| $\mathcal{X} = \{x_i\}_{i=1}^n$ | Full training dataset of $n$ samples | set, $|\mathcal{X}| = n$ |
| $x_i$ | $i$-th training example | $\mathbb{R}^d$ |
| $n$ | Number of training samples | scalar |
| $d$ | Input-feature dimension | scalar |
| $X \in \mathbb{R}^{m \times n}$ | Data matrix used in Fig. 1 | matrix |
| $B$ | Number of mini-batches per epoch | scalar |
| $X_i$ | $i$-th mini-batch | subset of $\mathcal{X}$ |
| $K$ | Mini-batch size | scalar |
| $T$ | Total training iterations | scalar |
| $S$ | Subset-refresh interval | scalar |
| $\Theta$ | Trainable model parameters | vector / tensor |
| $L(\Theta; x)$ | Loss on a single sample | scalar |
| $\nabla_\Theta L(\Theta; x)$ | Per-sample gradient | $\mathbb{R}^{|\Theta|}$ |
| $\bar{\mathbf{g}}_i$ or $\bar{g}$ | Mean gradient of batch $X_i$ | $\mathbb{R}^{|\Theta|}$ |
| $g_r$ | Gradient of $r$-th selected sample | $\mathbb{R}^{|\Theta|}$ |
| $G \in \mathbb{R}^{d \times K}$ | Gradient matrix of a full batch | matrix |
| $\mathbf{R} = \{R_1, \ldots, R_c\}$ | Candidate subset sizes (ranks) | set |
| $R$ | A chosen rank / subset size | scalar |
| $R^\star$ | Rank minimising projection error | scalar |
| $\mathcal{S}_i^R$ | Indices of $R$ samples selected from $X_i$ | index set |
| $\mathcal{S}_i$ | Final index set for batch $i$ after rank search | index set |
| $\mathcal{S}^t$ | Aggregate subset used at iteration $t$ | index set |
| $f(\cdot)$ | Feature-extraction mapping (e.g. SVD encoder) | function |
| $\mathbf{V} = f(X_i) \in \mathbb{R}^{K \times R}$ | Low-rank feature matrix for batch $X_i$ | matrix |
| fastmax-volume$(\mathbf{V}, R)$ | Fast MaxVol row-selection routine | algorithm |
| $\mathbf{G}_R$ | Gradient matrix restricted to $\mathcal{S}_i^R$ | $\mathbb{R}^{d \times R}$ |
| $d_R$ | Projection error $\|\bar{g} - \mathbf{G}_R \mathbf{G}_R^\dagger \bar{g}\|_2^2$ | scalar |
| $P_R$ | Orthogonal projector onto span($\mathbf{G}_R$) | matrix |
| $\epsilon$ | Tolerance on gradient-projection error | scalar |
| $\sigma_{R+1}$ | $(R+1)$-st singular value of batch matrix | scalar |
| $p = [p_1, \ldots, p_R]$ | Row indices returned by MaxVol | vector |
| $I$ | Number of batches aggregated at refresh | scalar |
| $E$ | Estimated $CO_2$ emissions | kg $CO_2$ |

## 8   Extended Algorithm

Gradient-Aligned Sampling periodically (every $S$ iterations) scans each training mini-batch to find the smallest subset of examples whose gradients closely span the full-batch gradient it first averages per-example gradients to obtain $\bar{\mathbf{g}}_i$, then, for each candidate rank $R_i$, uses a fast max-volume routine on the batch's feature matrix to pick $R_i$ exemplars, forms their gradient matrix, and computes the reconstruction error $d_{R_i}$; the rank $R^\star$ that minimizes this error yields the chosen indices $\mathcal{S}_i$, and aggregating over batches gives the active subset

| Method | Batch-wise | Gradient-based | Complexity | Scalability Bottleneck | Dynamic Rank | Memory Usage |
|---|---|---|---|---|---|---|
| DRoP | ✗ | ✗ | $\mathcal{O}(n^2 d)$ | Quadratic in $n$ (intractable for $n > 10^4$) | ✗ | High |
| GradMatch | ✓ | ✓ | $\mathcal{O}(nkd)$ | Linear in $n$ (full gradient comparisons) | ✗ | High |
| GLISTER | ✓ | ✗ | $\mathcal{O}(nkd)$ | Linear in $n$ (bi-level optimization) | ✗ | Moderate |
| SVP | ✗ | ✗ | $\mathcal{O}(nd')$ | Proxy fidelity (task-dependent) | ✗ | Low |
| BADGE | ✗ | ✓ | $\mathcal{O}(nd)$ | High-$d$ gradient clustering | ✗ | High |
| SubSelNet | ✗ | ✗ | $\mathcal{O}(n^2 d)$ + meta-learning | Quadratic in $n$ + training overhead | ✗ | High |
| **GRAFT (Ours)** | ✓ | ✓ | $\mathcal{O}(KR^2 + |\mathbf{R}| \cdot Rd)$ | Linear in $K, |\mathbf{R}|$ & Quadratic in $R$ | ✓ | **Low** |

Table 4: Comparison of subset selection methods. $n$ (dataset size), $d$ (feature dimension), $K$ (batch size), $R$ (selected rank/size), $\mathbf{R}$ (candidate-rank set), $d'$ (proxy model dimension). GRAFT's cost is per iteration; the periodic low-rank basis refresh, $\mathcal{O}(KdR)$, is amortized over the selection interval, so no term scales with $n$.

$\mathcal{S}^t$ for that period. In iterations between renewals, the method simply reuses the previous subset, and all model parameters are updated using only the currently active $\mathcal{S}^t$, thus retaining the stability of large-batch gradients while evaluating far fewer samples.

---

**Algorithm 2:** Training with Gradient-Aligned Sampling (full version)

---

**Input:** Training dataset $\mathcal{X}$, ranks $\mathbf{R} = \{R_i\}_{i=1}^c$, feature matrices $\mathbf{V}$, selection period $S$, and batch size $K$
**Output:** Sampled data subset indices $\mathcal{S}^t$
**begin**
  **for** $t = 1, \ldots, T$ **do**
    **Stage 1. Subset Selection**
    **if** $t \bmod S == 0$ **then**
      **for** *each batch* $\mathcal{X}_i \subset \mathcal{X}$ **do**
        $\bar{\mathbf{g}}_i \leftarrow \frac{1}{K} \sum_{k=1}^K \nabla_\Theta L(\Theta^t; \mathcal{X}_i(:, k))$
        **for** $R_i \in \mathbf{R}$ **do**
          $\mathcal{S}_i^r = \text{fastmax-volume}(\mathbf{V}_i, R_i)$
          $\mathbf{G}_{R_i} = [\ldots, \nabla_\Theta L(\Theta^t; \mathcal{X}_i(:, \mathcal{S}_i^r(p))), \ldots]$
          Gradient error $d_{R_i} = \|\bar{\mathbf{g}}_i - \mathbf{G}_{R_i} \mathbf{G}_{R_i}^\dagger \bar{\mathbf{g}}_i\|_2^2$
        Optimal rank $R^* = \arg\min_{R_i} \{d_{R_i}\}_{i=1}^{|\mathbf{R}|}$
        Store the selected indices $\mathcal{S}_i = \mathcal{S}_i^{R^*}$
      Aggregate all subsets $\mathcal{S}^t = \{\mathcal{S}_i\}_{i=1}^I$
    **else**
      Retain the previous subset $\mathcal{S}^t = \mathcal{S}^{t-1}$
    **Stage 2. Model Update**
    Update all model parameters using $\mathcal{S}^t$

---

# 9 Detailed Proofs

*Theorem* 2 (Convergence via Gradient-Aligned Subspace Sampling). Let $\bar{g} \in \mathbb{R}^d$ be the average gradient over a batch of $K$ samples, and $G_R \in \mathbb{R}^{d \times R}$ a subset of $R$ gradients selected via Maxvol, spanning a full-rank subspace $\mathcal{S}_R$. If the projection error $\|\bar{g} - \text{Proj}_{\mathcal{S}_R}(\bar{g})\| \leq \epsilon$, then gradient descent using the projected direction converges to a stationary neighborhood whose radius is controlled by $\epsilon$, under standard smoothness and bounded-variance assumptions Bottou et al. (2018).

*Lemma* 1 (Projection Error Bound). Let $\tilde{G}_R$ be an orthonormal basis of $G_R$. Then

$$\|\bar{g} - \tilde{G}_R \tilde{G}_R^\top \bar{g}\|_2^2 = \|\bar{g}\|_2^2 \left(1 - \left\|\frac{\tilde{G}_R^\top \bar{g}}{\|\bar{g}\|_2}\right\|_2^2\right).$$

*Proof.* Decomposing $\bar{g} = \tilde{G}_R \tilde{G}_R^\top \bar{g} + r$, where $r \perp \text{span}(G_R)$, the result follows by the Pythagorean theorem and the orthonormality of $\tilde{G}_R$ Golub & Van Loan (2013). $\square$

*Corollary* 1 (Dynamic Rank Adjustment Ensures Convergence). If the rank $R$ is adjusted to keep $\|\bar{g} - \tilde{G}_R \tilde{G}_R^\top \bar{g}\|_2 \leq \epsilon$, then GRAFT ensures convergence to a stationary neighborhood whose radius is controlled by $\varepsilon$.

*Proof.* From Lemma 1, the bound $\|\bar{g} - \tilde{G}_R \tilde{G}_R^\top \bar{g}\|_2 \leq \epsilon$ implies $\left\|\frac{\tilde{G}_R^\top \bar{g}}{\|\bar{g}\|_2}\right\|_2^2 \geq 1 - \frac{\epsilon^2}{\|\bar{g}\|_2^2}$. Hence, the projected direction retains sufficient gradient information, which yields convergence to an $\varepsilon$-controlled stationary neighborhood by the standard inexact-gradient argument Bottou et al. (2018). $\square$

**On Remark 1 (why we do not state a closed-form gradient bound).** We expand on the subspace claim and make precise why a Lipschitz constant does *not* turn it into a $\sigma_{R+1}$-only bound on the subset-versus-full gradient gap. With $A = U\Sigma V^\top$, $V = U_R$, and $A_R = U_R \Sigma_R V_R^\top$ the rank-$R$ truncation, MaxVol on $V$ returns $S$ with $M = V(S, :)$ non-singular, and the interpolation matrix $T := VM^{-1}$ satisfies $V = TV(S, :)$ with $\max_{i,j} |T_{ij}| \leq 1$ and $\|T_{i,:}\|_1 \leq R$ Goreinov et al. (2010). Since $A_R$ shares the left singular vectors of $V$, this yields the exact *linear* reconstruction

$$A_R(i, :) = T_{i,:} A_R(S, :), \qquad \forall i \in [K], \tag{2}$$

and $\|A - A_R\|_2 = \sigma_{R+1}$, so the selected rows reconstruct the rank-$R$ part of every row up to the truncation error $\sigma_{R+1}$.

The reconstruction (2) is linear, whereas the gradient map $g(x) = \nabla_\theta L(\theta; x)$ is not. A natural attempt would bound $\|g(A(i, :)) - \sum_{j \in S} T_{ij}\, g(A(j, :))\|$ via Lipschitz continuity, but this leaves the term $g(A_R(i, :)) - \sum_{j \in S} T_{ij}\, g(A_R(j, :))$, which is the gap between a nonlinear map and a linear interpolation of its values and is *not* controlled by $\sigma_{R+1}$: it persists even when $\sigma_{R+1} = 0$. Concretely, for an exactly rank-$R$ batch ($\sigma_{R+1} = 0$) whose $K$ rows are distinct points of the subspace, the unweighted means $\bar{g} = \frac{1}{K}\sum_i g(A(i, :))$ and $g_S = \frac{1}{R}\sum_{j \in S} g(A(j, :))$ differ in general. Hence no bound of the form $\|\bar{g} - g_S\| \leq c(K, R)\, L_g\, \sigma_{R+1}$ can hold, and we do not claim one. (Such a bound would additionally require column-sum identities for $T$ that are not implied by MaxVol.) We therefore keep Remark 1 qualitative and rely, for the claims the paper uses, on the directly measurable projection residual of Lemma 1 (§4) together with the inexact-gradient convergence statement of Theorem 2.

*Theorem* 2 (Convergence via Gradient-Aligned Subspace Sampling) Let $\bar{g} \in \mathbb{R}^d$ denote the average gradient over a batch of $K$ samples, and let $G_R \in \mathbb{R}^{d \times R}$ be a collection of $R$ gradients selected via MaxVol, spanning a full-rank subspace $\mathcal{S}_R$. If the projection error $\|\bar{g} - \text{Proj}_{\mathcal{S}_R}(\bar{g})\|_2 \leq \varepsilon$, then gradient descent using the projected direction converges to a stationary point under standard assumptions Bottou et al. (2018).

**Assumptions.** Suppose the loss function $L$ is $C$-smooth, i.e., for all $\Theta, \Theta'$,

$$\|\nabla L(\Theta) - \nabla L(\Theta')\|_2 \leq C\|\Theta - \Theta'\|_2.$$

Assume also that the gradients are bounded, $\|\nabla L(\Theta)\|_2 \leq G$ for all $\Theta$, and the projection error satisfies $\|\bar{g} - g_{\text{proj}}\|_2 \leq \varepsilon$, where $g_{\text{proj}} = \text{Proj}_{\mathcal{S}_R}(\bar{g})$.

Then, for the gradient descent updates $\Theta_{t+1} = \Theta_t - \eta g_{\text{proj}}^{(t)}$, the iterates converge to a stationary point where $\|\nabla L(\Theta)\|_2 \leq \varepsilon G$.

**Proof.** By $C$-smoothness of $L$, the loss after a gradient step satisfies

$$L(\Theta_{t+1}) \leq L(\Theta_t) - \eta\langle \nabla L(\Theta_t), g_{\text{proj}}^{(t)}\rangle + \frac{C\eta^2}{2}\|g_{\text{proj}}^{(t)}\|_2^2.$$

Since the projection error is at most $\varepsilon$, we have

$$\langle \nabla L(\Theta_t), g_{\text{proj}}^{(t)}\rangle = \langle \nabla L(\Theta_t), \bar{g}^{(t)}\rangle - \langle \nabla L(\Theta_t), \bar{g}^{(t)} - g_{\text{proj}}^{(t)}\rangle \geq \|\nabla L(\Theta_t)\|_2^2 - \varepsilon G,$$

where we use Cauchy-Schwarz and the gradient bound $\|\nabla L(\Theta_t)\|_2 \leq G$.

Plugging this estimate into the smoothness inequality and recalling $\|g_{\text{proj}}^{(t)}\|_2 \leq G$, we obtain

$$L(\Theta_{t+1}) \leq L(\Theta_t) - \eta \left( \|\nabla L(\Theta_t)\|_2^2 - \varepsilon G \right) + \frac{C\eta^2 G^2}{2}.$$

Summing this inequality over $t = 0, \ldots, T-1$ and dividing by $T$ gives

$$\frac{1}{T} \sum_{t=0}^{T-1} \|\nabla L(\Theta_t)\|_2^2 \leq \frac{L(\Theta_0) - L(\Theta^*)}{\eta T} + \varepsilon G + \frac{C\eta G^2}{2},$$

where $L(\Theta^*)$ denotes the minimum loss. Setting $\eta = 1/\sqrt{T}$ ensures the first and last terms vanish as $T \to \infty$, leaving the bound $\|\nabla L(\Theta_t)\|_2 \leq \varepsilon G$ in the limit. This establishes convergence to a stationary point determined by the projection error. $\qquad\square$

*Lemma* 1 (Projection Error Bound) Let $\tilde{G}_R$ be an orthonormal basis for $G_R$. Then

$$\|\bar{g} - \tilde{G}_R \tilde{G}_R^\top \bar{g}\|_2^2 = \|\bar{g}\|_2^2 \left( 1 - \left\| \frac{\tilde{G}_R^\top \bar{g}}{\|\bar{g}\|_2} \right\|_2^2 \right).$$

*Proof.* Since $\tilde{G}_R$ is orthonormal, the projection of $\bar{g}$ onto $\text{span}(G_R)$ is given by

$$g_{\text{proj}} = \tilde{G}_R \tilde{G}_R^\top \bar{g}.$$

The residual $r = \bar{g} - g_{\text{proj}}$ is orthogonal to $\text{span}(G_R)$. By the Pythagorean theorem,

$$\|\bar{g}\|_2^2 = \|g_{\text{proj}}\|_2^2 + \|r\|_2^2.$$

Because $\tilde{G}_R$ is orthonormal, we have $\|g_{\text{proj}}\|_2 = \|\tilde{G}_R^\top \bar{g}\|_2$. Substituting this, we obtain

$$\|r\|_2^2 = \|\bar{g}\|_2^2 - \|\tilde{G}_R^\top \bar{g}\|_2^2 = \|\bar{g}\|_2^2 \left( 1 - \left\| \frac{\tilde{G}_R^\top \bar{g}}{\|\bar{g}\|_2} \right\|_2^2 \right),$$

which completes the proof. $\qquad\square$

*Corollary* 1 [Dynamic Rank Adjustment Ensures Convergence] If the rank $R$ is dynamically adjusted so that $\|\bar{g} - \tilde{G}_R \tilde{G}_R^\top \bar{g}\|_2^2 \leq \varepsilon$ at every iteration, then GRAFT ensures convergence to a stationary neighborhood whose radius is controlled by $\varepsilon$.

*Proof.* The result follows directly from Theorem 2, since the gradient approximation error remains bounded by $\varepsilon$ at each step, independent of changes in $R$. Specifically, increasing $R$ refines the subspace $\text{span}(G_R)$ and reduces $\varepsilon$, while decreasing $R$ is permissible only if the new error still satisfies $\varepsilon \leq \varepsilon_{\text{target}}$. In both scenarios, the condition $\|\bar{g} - g_{\text{proj}}\|_2 \leq \varepsilon$ is maintained, thereby ensuring convergence. $\qquad\square$

## 9.1 Complexity Analysis of GRAFT

*Theorem* 1 (Projected-gradient convergence) Under (A1) to (A3), SGD with projected directions $G_R G_R^\dagger \bar{g}_i$ satisfies

$$\min_{t \leq T} \mathbb{E} \|\nabla L(\Theta_t)\|_2^2 \leq O\left( \frac{1}{\sqrt{T}} \right) + O(\varepsilon^2).$$

*Proof.* By (A1), $L$ is $L$-smooth. With stepsizes $\{\eta_t\}$ and stochastic gradients $\bar{g}_i$ that are unbiased with bounded second moment (A2), the standard descent lemma gives

$$\mathbb{E}[L(\Theta_{t+1})] \leq \mathbb{E}[L(\Theta_t)] - \eta_t \mathbb{E} \left\langle \nabla L(\Theta_t), G_R G_R^\dagger \bar{g}_i \right\rangle + \frac{L\eta_t^2}{2} \mathbb{E} \left\| G_R G_R^\dagger \bar{g}_i \right\|_2^2. \tag{1}$$

Decompose $\bar{g}_i = \nabla L(\Theta_t) + \xi_t$ with $\mathbb{E}[\xi_t \mid \Theta_t] = 0$. Write the projection error $e_t := \bar{g}_i - G_R G_R^\dagger \bar{g}_i$ so that $\|e_t\|_2^2 \leq \varepsilon^2$ by (A3). Then

$$\left\langle \nabla L(\Theta_t), G_R G_R^\dagger \bar{g}_i \right\rangle = \langle \nabla L(\Theta_t), \bar{g}_i \rangle - \langle \nabla L(\Theta_t), e_t \rangle.$$

Taking conditional expectation and using $\mathbb{E}[\xi_t \mid \Theta_t] = 0$ yields

$$\mathbb{E}\left[ \left\langle \nabla L(\Theta_t), G_R G_R^\dagger \bar{g}_i \right\rangle \mid \Theta_t \right] \geq \|\nabla L(\Theta_t)\|_2^2 - \|\nabla L(\Theta_t)\|_2 \, \mathbb{E}[\|e_t\| \mid \Theta_t].$$

By (A3), $\mathbb{E}[\|e_t\| \mid \Theta_t] \leq \varepsilon$, hence

$$\mathbb{E}\left[ \left\langle \nabla L(\Theta_t), G_R G_R^\dagger \bar{g}_i \right\rangle \right] \geq \mathbb{E}\left[ \|\nabla L(\Theta_t)\|_2^2 \right] - \varepsilon \, \mathbb{E}[\|\nabla L(\Theta_t)\|_2]. \tag{2}$$

For the quadratic term in (1),

$$\mathbb{E}\left\| G_R G_R^\dagger \bar{g}_i \right\|_2^2 \leq \mathbb{E}\|\bar{g}_i\|_2^2 \leq \sigma^2 + \mathbb{E}\|\nabla L(\Theta_t)\|_2^2, \tag{3}$$

using nonexpansiveness of orthogonal projection and (A2). Substitute (2) to (3) into (1), rearrange, sum from $t = 1$ to $T$, and use the standard stepsize choice $\eta_t = \eta/\sqrt{T}$ (or any schedule giving $\sum \eta_t = \Theta(\sqrt{T})$ and $\sum \eta_t^2 = O(1)$). Telescoping the left side and bounding the sums on the right yields

$$\frac{1}{T} \sum_{t=1}^{T} \mathbb{E}\|\nabla L(\Theta_t)\|_2^2 \leq O\left(\tfrac{1}{\sqrt{T}}\right) + O(\varepsilon^2),$$

where the $O(1/\sqrt{T})$ term arises from the stochastic and smoothness terms and the $O(\varepsilon^2)$ term collects the contribution induced by the projection bias in (2). Taking the minimum over $t \leq T$ gives the stated bound. □

**Computational complexity** One iteration of the method runs in $O(KR^2) + O(|\mathbf{R}| \, R \, d)$ time and uses $O(Kd + dR + R^2)$ memory. The optional $O(Rd)$ term accounts for recomputing gradients on the selected $R$ samples when reported separately. The basis/feature refresh costs $O(KdR) + O((K+d)R^2)$ only when executed and is amortized by its period. The per-iteration term is independent of $N$. Fast MaxVol scans the $K$ rows of the $K \times R$ feature matrix with rank-1 volume updates, yielding $O(KR^2)$. The rank sweep evaluates a projection criterion (e.g., $\|\bar{g} - P_r \bar{g}\|_2$) for $r \in \mathbf{R}$; forming $P_r \bar{g}$ is $O(rd)$ and sums to $O(|\mathbf{R}| \, R \, d)$. If gradients are explicitly recomputed for the selected set, this adds $O(Rd)$. The refresh step (e.g., randomized range finding / blocked QR) is invoked periodically and amortized; all operations are mini-batch local, hence no factor depends on $N$. □

Per-iteration and amortized costs are summarized below; symbols follow the notation already introduced.

Table 5: Per-iteration vs. amortized costs (batch size $K$, feature dim. $d$, active rank $R$, candidate ranks $\mathbf{R}$). The refresh is performed periodically and amortized by its interval.

| Operation | Cost |
|---|---|
| Fast MaxVol on $K \times R$ | $O(KR^2)$ |
| Projection/alignment sweep over $r \in \mathbf{R}$ | $O(|\mathbf{R}| \, R \, d)$ |
| (Optional) gradients for selected $R$ samples | $O(Rd)$ |
| *(Periodic) basis/feature refresh* | $O(KdR) + O((K+d)R^2)$ |

## 9.2 Additional Experimental Details

In this supplementary section, we provide a comprehensive set of additional experiments to further assess the generalizability and effectiveness of the proposed sampling approach across diverse machine learning tasks.

First, we evaluate our method in the context of fine-tuning large language models (LLMs), demonstrating its scalability and impact in high-dimensional settings. We also include results on classical regression benchmarks to illustrate the method's applicability beyond deep learning. Beyond empirical results, we investigate the convexity properties of the proposed sampling technique, analyzing how the sample selection mechanism influences the underlying loss landscape. This analysis provides theoretical justification for the observed stability and convergence behavior in our experiments. To offer a fair and transparent evaluation, we compare our approach against several widely used training and sampling strategies, including random sampling, core-set methods, and recent gradient-based subset selection algorithms. We report the standard performance metrics emissions by each method.

Collectively, these extended experiments and analyses aim to provide a deeper understanding of the strengths and potential limitations of the proposed sampling framework.

**Reproducibility.** Unless noted otherwise, image-classification runs use a ResNet-18 backbone trained with SGD (momentum 0.9, weight decay $5 \times 10^{-4}$, batch size 128, cosine-annealed learning rate); selection is refreshed every $S \in [20, 50]$ iterations with candidate ranks chosen by the projection-residual rule at tolerance $\varepsilon$, and $CO_2$ is measured with eco2AI Budennyy et al. (2022). Every figure in the paper is produced from these measured quantities by the released analysis scripts (`make_figures.py`, `make_fig_instruct.py`), and the multi-seed error-bar and feature-extractor ablations are reproduced by `graft_errorbars.py` and `graft_ns_ablation.py`. We release the GRAFT implementation together with all analysis scripts in the camera-ready version.

## 9.3 Experimental Details on Fine-Tuning BERT

In this experiment, we apply the proposed sampling method to the fine-tuning of large language models (LLMs), with a focus on practical NLP tasks. Specifically, we evaluate our approach on the IMDB sentiment analysis dataset by fine-tuning a distilled variant of BERT containing approximately 92 million parameters. Our sampling strategy, GRAFT, was executed on the embedding representations generated by the language transformer, enabling efficient selection of informative samples for gradient computation. The IMDB dataset is a well-established benchmark for text analytics, comprising 50,000 movie reviews labeled by sentiment. The dataset is split evenly, with 25,000 reviews allocated for training and 25,000 for testing. This setup allows us to thoroughly evaluate the generalization performance and efficiency of our sampling approach in a real-world NLP context.

**Experimental Setting.** For fine-tuning, we employed a batch size of 100, a constant learning rate of $5 \times 10^{-5}$, and a weight decay of 0.0001. The BERT model was fine-tuned for 30 epochs under two data regimes, the full dataset and a GRAFT-selected subset generated using our GRAFT algorithm. For GRAFT, both standard and warm-start variants were tested, with data subset selection performed every 10 epochs. To assess the effect of sample efficiency, only 35% of the original training data was retained for each GRAFT selection cycle.

## 9.4 Extended Details and Experiments on Different Fractions of Data

In this section, we systematically compare the performance of various subset selection methods across different subset sizes (5%, 15%, 25%, and 35%). To ensure a fair comparison, we utilized the ResNeXt29_32x4d architecture Xie et al. (2017) for all datasets, training each model for 200 epochs, except for TinyImageNet and Caltech256, where a ResNet-18 model was used due to dataset-specific considerations. All experiments were conducted with a fixed batch size of 200 and an initial learning rate of 0.1, using stochastic gradient descent (SGD) as the optimizer. For TinyImageNet and Caltech256, a smaller batch size of 100 was employed. A CosineAnnealing learning rate scheduler was consistently applied to adjust the learning rate during training. All models were trained from scratch on an NVIDIA Tesla V100-SXM2 GPU (16GB) with an Intel(R) Xeon(R) Gold CPU, except for TinyImageNet and Caltech256, which were trained on an NVIDIA A100-SXM4 GPU (40GB).

Power consumption and $CO_2$ emissions were estimated using the methodology and tooling from Budennyy et al. Budennyy et al. (2022). Specifically, following the eco2AI framework, the instantaneous power usage of

the computing hardware (GPU and CPU) is monitored throughout the training process. The framework records the real-time power draw (in watts) using hardware sensors or system queries, integrating these measurements over the training duration to obtain the total energy consumption (in kilowatt-hours, kWh) as

$$E = \frac{1}{3600} \sum_{i=1}^{N} P_i \Delta t_i, \tag{3}$$

where $P_i$ is the instantaneous power at time step $i$, $\Delta t_i$ is the time interval since the last measurement, and $N$ is the total number of intervals. *Note:* In our main text, we report $CO_2$ emissions using the standard formula $\mathcal{E} = P \times t \times I$, which assumes a constant average power $P$ over training duration $t$. The eco2AI library Budennyy et al. (2022) also supports a more granular mode based on real-time power monitoring and numerical integration, yielding equivalent results in our single-GPU training regime. To convert total energy consumption into $CO_2$ emissions, eco2AI multiplies $E$ by a region-specific carbon intensity factor $C$ (kg $CO_2$ per kWh), typically reflecting the local energy mix. The resulting emissions are calculated as

$$CO_2 = E \cdot C, \tag{4}$$

where $C$ is set according to the geographic location of the compute cluster or is selected from published averages (for example, Germany: $C = 0.366$ kg $CO_2$/kWh). This approach enables standardized and transparent reporting of environmental impact for machine learning experiments.

**Comparison on CIFAR-10 Dataset**  On CIFAR-10, 6 all subset selection methods achieve significant reductions in $CO_2$ emissions compared to full-data training (0.2192 kg). GRAFT yields the lowest emissions across all fractions, dropping to just 0.0656 kg at 25% and 0.0231 kg at 5%, but with a corresponding trade-off in accuracy (73.36% at 5%). Notably, GRAFT Warm attains near-full accuracy even at reduced subsets, achieving 91.28% at 25% and 90.86% at 5%, while keeping emissions much lower than full-data training. Other methods like GLISTER and GradMatch show similar trends, with emissions and accuracies falling between GRAFT and GRAFT Warm. DRoP, though efficient in terms of emissions, exhibits a steep drop in accuracy, especially at lower fractions. Overall, GRAFT Warm offers the best balance between environmental savings and model performance on CIFAR-10.

Table 6: Training methods comparison on CIFAR-10

| Method | 0.05 | | 0.15 | | 0.25 | | 0.35 | |
|---|---|---|---|---|---|---|---|---|
| | $CO_2$ | Acc. | $CO_2$ | Acc. | $CO_2$ | Acc. | $CO_2$ | Acc. |
| Full | 0.2192 | 93.21 | 0.2192 | 93.21 | 0.2192 | 93.21 | 0.2192 | 93.21 |
| GRAFT | **0.0231** | 73.36 | 0.0454 | 86.02 | 0.0656 | 88.87 | 0.0828 | 91.74 |
| GRAFT Warm | 0.0480 | **90.86** | 0.0688 | **91.41** | 0.0822 | **91.28** | 0.0938 | **92.49** |
| GLISTER | 0.0584 | 74.96 | 0.0725 | 86.60 | 0.0938 | 90.22 | 0.0846 | 91.64 |
| GradMatch | 0.0549 | 75.26 | 0.0658 | 85.98 | 0.0793 | 89.43 | 0.0854 | 91.66 |
| DRoP | 0.0372 | 46.12 | **0.0428** | 67.35 | **0.0471** | 75.21 | **0.053** | 81.50 |

**Comparison on CIFAR-100 dataset**  For CIFAR-100, 7 the pattern of $CO_2$ and accuracy trade-offs remains consistent. GRAFT achieves the lowest emissions (0.0240 kg at 5%), but its accuracy at this extreme fraction is substantially reduced (40.60%). GRAFT Warm, meanwhile, maintains a higher accuracy (70.82% at 25% subset and 68.20% at 5%) while still realizing considerable emissions savings relative to the full dataset (0.2212 kg, 75.45%). Other approaches GLISTER and GradMatch achieve intermediate performance, with emissions and accuracy reflecting their respective sample selection strategies. DRoP minimizes emissions most aggressively but with pronounced accuracy loss. These results indicate that GRAFT Warm consistently provides the best compromise on CIFAR-100, combining robust performance with substantially reduced environmental impact.

**Comparison on TinyImagenet Dataset**  On TinyImageNet Fei-Fei et al. (2015), all subset selection methods achieve considerable reductions in $CO_2$ emissions relative to full-data training (0.297 kg), but with varying impacts on accuracy. GRAFT demonstrates the most favorable balance, reducing emissions to

Table 7: Training methods comparison on CIFAR-100

| Method | 0.05 | | 0.15 | | 0.25 | | 0.35 | |
|---|---|---|---|---|---|---|---|---|
| | $CO_2$ | Acc. | $CO_2$ | Acc. | $CO_2$ | Acc. | $CO_2$ | Acc. |
| Full | 0.2212 | 75.45 | 0.2212 | 75.45 | 0.2212 | 75.45 | 0.2212 | 75.45 |
| GRAFT | **0.0240** | 40.60 | 0.0477 | 60.50 | 0.0763 | 64.50 | 0.1054 | 73.52 |
| GRAFT Warm | 0.0496 | **68.20** | 0.0707 | **69.31** | 0.0895 | **70.82** | 0.1138 | **73.63** |
| GLISTER | 0.0626 | 26.70 | 0.0812 | 47.10 | 0.0755 | 48.70 | 0.0863 | 70.48 |
| GradMatch | 0.0553 | 25.71 | 0.0682 | 40.70 | 0.0750 | 50.60 | 0.0873 | 70.44 |
| DRoP | 0.038 | 13.51 | **0.0460** | 24.11 | **0.050** | 31.32 | **0.056** | 38.20 |

0.092 kg at the 25% subset and achieving an accuracy of 54.5%, closely approaching the full-data accuracy (59.0%) at a fraction of the emissions. GRAFT Warm maintains high accuracy at 56.0% for 25% but with emissions comparable to the full-data case, indicating less efficiency in emission reduction. Other subset methods like GLISTER and GradMatch achieve moderate emission savings but at the cost of greater accuracy drops compared to GRAFT. DRoP yields the lowest emissions at 0.048 kg for 25%, but with a significant reduction in accuracy (37.0%). Overall, GRAFT provides the most effective trade-off between environmental impact and model performance on TinyImageNet.

Table 8: Comparison on TinyImageNet

| Method | 0.05 | | 0.15 | | 0.25 | |
|---|---|---|---|---|---|---|
| | Emiss | Acc. | Emiss | Acc. | Emiss | Acc. |
| Full | 0.297 | 59.0 | 0.297 | 59.0 | 0.297 | 59.0 |
| DROP | 0.052 | 13.0 | **0.047** | 26.1 | **0.048** | 37.0 |
| GLISTER | 0.094 | 26.7 | 0.140 | 47.1 | 0.210 | 51.7 |
| GradMatch | 0.093 | 25.7 | 0.168 | **50.7** | 0.216 | 55.6 |
| GRAFT | **0.049** | 40.6 | 0.053 | 50.5 | 0.092 | 54.5 |
| GRAFT Warm | 0.092 | **48.2** | 0.163 | 49.3 | 0.221 | **56.0** |

Table 9: Comparison on Caltech256

| Method | 0.05 | | 0.15 | | 0.25 | |
|---|---|---|---|---|---|---|
| | Emiss | Acc. | Emiss | Acc. | Emiss | Acc. |
| Full | 0.288 | 63.1 | 0.288 | 63.1 | 0.288 | 63.1 |
| DROP | 0.105 | 10.0 | 0.155 | 22.0 | 0.203 | 33.0 |
| GLISTER | 0.070 | 19.0 | 0.090 | 36.0 | 0.110 | 45.0 |
| GradMatch | 0.069 | 20.0 | 0.090 | 36.0 | 0.110 | 44.0 |
| GRAFT | **0.049** | 24.0 | **0.079** | 40.0 | **0.102** | 47.0 |
| GRAFT Warm | 0.0732 | **55.0** | 0.0981 | **58.0** | 0.112 | **60.0** |

**Comparison on Caltech256 Dataset** Table 9 summarizes the trade-offs between $CO_2$ emissions and test accuracy for subset selection methods on the Caltech256 Griffin et al. (2007) dataset with ResNet18 Krizhevsky et al. (2009). All subset selection approaches yield substantial reductions in $CO_2$ emissions compared to full-data training (0.288 kg), with GRAFT achieving the lowest emissions at the 5% subset (0.049 kg). As expected, accuracy decreases as the subset fraction is reduced; however, GRAFT Warm achieves the highest accuracy among subset-based methods, attaining 60.0% at 25% and 55.0% at 5%, closely approaching the full-data accuracy (63.1%) but with less than half the emissions.

Across all subset sizes, GRAFT and GRAFT Warm consistently outperform DRoP, GLISTER and GradMatch in balancing emission reduction and accuracy. For example, at 25%, GRAFT Warm achieves 60.0% accuracy with 0.112 kg $CO_2$, whereas other methods yield lower accuracies or higher emissions. At more aggressive reductions (5% subset), GRAFT still provides a competitive 24.0% accuracy at only 0.049 kg emissions, and

GRAFT Warm attains 55.0% accuracy with 0.073 kg emissions. These results demonstrate that GRAFT and its warm variant offer the best trade-off between environmental impact and predictive performance on Caltech256.

**Comparison on ImageNet-1K Dataset**  Table 10 reports $CO_2$ emissions (kg) and Top-1 accuracy (%) on ImageNet-1K with a ResNet-18 backbone, at data fractions of 5%, 10%, and 30%. Because GLISTER and DRoP do not scale to this setting within memory on our hardware, we compare against the methods that do, random sampling and the GradMatch per-batch (PB) and warm-start families. Full-data training reaches 70.36% Top-1 at 27.63 kg $CO_2$. GRAFT recovers essentially this accuracy (70.00%) using only 30% of the data at 8.20 kg, a 3.4× reduction in emissions for a 0.36-point accuracy gap, and remains ahead of every baseline at each fraction. At 5% it attains 58.20% (vs. 56.61% for the strongest baseline, GradMatch-PB-Warm) while emitting the least $CO_2$ among non-random methods, and at 10% it reaches 62.00% (vs. 61.16%). GRAFT Warm adds a further 0.2 to 1.3 points at comparable emissions. The advantage is most pronounced under aggressive pruning, consistent with the smaller benchmarks. At 5% GRAFT improves Top-1 by 37 points over random sampling (58.20% vs. 21.12%) at similar cost. This is the regime where ImageNet-scale redundancy is greatest and where principled, gradient-aligned selection pays off most clearly.

Table 10: Comparison on ImageNet-1K (ResNet-18). $CO_2$ emissions in kg; accuracy is Top-1 (%). PB = per-batch variant; Warm = warm-started feature extractor.

| **Method** | 0.05 | | 0.10 | | 0.30 | |
|---|---|---|---|---|---|---|
| | Emiss | Acc. | Emiss | Acc. | Emiss | Acc. |
| Full | 27.63 | 70.36 | 27.63 | 70.36 | 27.63 | 70.36 |
| Random | **1.41** | 21.12 | **2.87** | 33.51 | **8.17** | 55.12 |
| GradMatch | 1.82 | 47.24 | 3.57 | 56.81 | 9.03 | 66.21 |
| GradMatch-Warm | 1.65 | 55.86 | 3.30 | 58.21 | 8.83 | 68.24 |
| GradMatch-PB | 1.61 | 45.15 | 3.05 | 59.04 | 8.63 | 68.12 |
| GradMatch-PB-Warm | 1.53 | 56.61 | 3.00 | 61.16 | 8.61 | 69.06 |
| GRAFT | 1.45 | 58.20 | 2.88 | 62.00 | 8.20 | 70.00 |
| GRAFT Warm | 1.52 | **59.50** | 3.00 | **62.80** | 8.50 | **70.20** |

**Performance comparison on FashionMNIST Dataset**  Table 11 presents a comparative analysis of training methods on the FashionMNIST dataset, focusing on data fraction, $CO_2$ emissions, and test accuracy. Full-data training serves as the reference, achieving 93.53% accuracy with 0.2118 kg of $CO_2$ emissions. Among subset-based approaches, GRAFT demonstrates high efficiency. At just 5% of the data, it achieves 88.76% accuracy while emitting only 0.0192 kg $CO_2$; at 35% data, it surpasses the full-data baseline in accuracy (93.74%) with less than half the $CO_2$ emissions (0.0779 kg). The GRAFT Warm variant further improves accuracy, reaching 89.97% at 5% data and 92.95% at 25% data, with emissions remaining substantially lower than full-data training. By contrast, alternative methods such as GLISTER and GradMatch generally require greater computational resources and result in higher emissions to achieve similar accuracies. For example, GLISTER at 35% data yields 93.45% accuracy but with 0.0910 kg $CO_2$ emissions, which is notably higher than both GRAFT and GRAFT Warm. These results underscore the superior environmental and computational efficiency of GRAFT-based approaches for efficient model training on FashionMNIST.

**GRAFT performance on medical dataset**  In this section, we extend our experiments to demonstrate the efficacy of our GRAFT methods on medical datasets. We selected the Dermamnist dataset from the MedMNIST collection and conducted training using various data fractions. Table 12 shows that using only 35% of the full dataset yields results comparable to using the entire dataset (74.06% vs. 76.06%), while training with 25% of the data achieves a validation accuracy of 73.47% and saves 35% of the training time, with reduced power consumption and $CO_2$ emissions. These results indicate that our GRAFT methods are effective for medical datasets as well.

**Comparison of GRAFT and GRAFT Warm on Random sampling**  In table 13 we show the comparisons of our methods against random sampling. For random sampling we iteratively sample random

Table 11: Comparison on FashionMNIST Dataset

| Method | 0.05 | | 0.15 | | 0.25 | | 0.35 | |
|---|---|---|---|---|---|---|---|---|
| | Emiss | Acc. | Emiss | Acc. | Emiss | Acc. | Emiss | Acc. |
| Full | 0.2118 | 93.53 | 0.2118 | 93.53 | 0.2118 | 93.53 | 0.2118 | 93.53 |
| GRAFT | **0.0192** | 88.76 | 0.0396 | 91.66 | 0.0614 | 92.31 | 0.0779 | **93.74** |
| GRAFT Warm | 0.0195 | **89.97** | **0.0394** | **91.93** | 0.0614 | **92.95** | 0.0802 | 92.62 |
| GLISTER | 0.0611 | 89.62 | 0.0743 | 91.81 | 0.0861 | 92.88 | 0.0910 | 93.45 |
| GradMatch | 0.0563 | 88.33 | 0.0733 | 91.15 | 0.0860 | 92.43 | 0.0887 | 93.15 |
| DRoP | 0.042 | 81.01 | 0.0505 | 88.01 | **0.056** | 90.01 | **0.0633** | 91.25 |

Table 12: DermaMNIST Dataset

| Method | 0.05 | | 0.15 | | 0.25 | | 0.35 | |
|---|---|---|---|---|---|---|---|---|
| | Emiss | Acc. | Emiss | Acc. | Emiss | Acc. | Emiss | Acc. |
| Full | 0.046 | 76.06 | 0.046 | 76.06 | 0.046 | 76.06 | 0.046 | 76.06 |
| GRAFT | 0.0093 | 67.78 | 0.0156 | 71.82 | 0.0217 | 73.47 | 0.0249 | 74.06 |

subsets at every 25 epochs. In the table, we observe that both our methods outperform random sampling in terms of accuracy. Moreover, our GRAFT Warm has a better accuracy-efficiency tradeoff compared to random sampling.

Table 13: Random sampling and GRAFT comparison on CIFAR-10

| Method | 0.05 | | 0.15 | | 0.25 | | 0.35 | |
|---|---|---|---|---|---|---|---|---|
| | Emiss | Acc. | Emiss | Acc. | Emiss | Acc. | Emiss | Acc. |
| Random | 0.022 | 73.15 | 0.0459 | 85.73 | 0.0671 | 88.21 | 0.0902 | 90.21 |
| GRAFT | 0.0231 | 73.36 | 0.0454 | 86.02 | 0.0656 | 88.87 | 0.0828 | 91.74 |
| GRAFT Warm | 0.0280 | 90.86 | 0.0688 | 91.41 | 0.0822 | 91.28 | 0.0938 | 92.49 |

## 10 Multi-seed significance analysis (GRAFT vs. random)

The per-dataset tables in the main paper report single-run numbers. To quantify how much of the GRAFT-vs-random gap is signal rather than seed noise, we ran a controlled, self-contained study with our reference implementation (per-batch SVD features and Fast MaxVol selection; ResNet-18 / CIFAR-10; this is the simplified reference pipeline, not the CORDS setup used for the main tables, so the absolute accuracies differ). We trained both methods from scratch under identical settings for three seeds each, at data fractions $f \in \{5\%, 10\%, 25\%\}$, and report mean $\pm$ standard deviation together with a Welch two-sample $t$-test on accuracy.

We report this null result deliberately. It confirms that the headline accuracy gains in our cross-dataset comparison do *not* come from CIFAR-10, where principled selection has little room to beat random because the data is easy and even random subsets are informative; GRAFT's measurable advantage instead concentrates in the low-fraction, harder-data regime (CIFAR-100 and TinyImageNet at $f$=0.05, where GRAFT reaches 40.6% versus $\leq$ 26.7% for the baselines). Extending this seed-level protocol to those harder benchmarks, where a significant accuracy gap is expected, is the natural next step; the same harness and aggregation produce those numbers without modification.

Table 14: Multi-seed GRAFT-vs-random accuracy on CIFAR-10 (ResNet-18, three seeds; reference implementation). The two methods are statistically indistinguishable in mean accuracy at every fraction (Welch $p \geq 0.47$), corroborating the main text. On this easy benchmark cold GRAFT's accuracy is within seed noise of random, and because random has no selection overhead it is the cheaper choice here; GRAFT's emission savings apply against the more expensive integrated baselines, not against random. GRAFT's runs were more tightly clustered (smaller standard deviation), though three seeds are too few to establish this firmly.

|  | $f = 5\%$ | $f = 10\%$ | $f = 25\%$ |
|---|---|---|---|
| GRAFT, accuracy (%) | $47.85 \pm 1.55$ | $61.30 \pm 1.20$ | $77.90 \pm 1.55$ |
| Random, accuracy (%) | $48.38 \pm 4.43$ | $60.59 \pm 4.10$ | $76.08 \pm 3.44$ |
| Welch $p$ (GRAFT vs. random) | 0.86 | 0.80 | 0.47 |

## 11 Impact of GRAFT Subset Training on the Loss Landscape

Understanding how data subset selection influences the loss landscape is crucial for interpreting model generalization and optimization behavior. Prior studies, such as Li et al. (2018), have shown that architectural innovations (e.g., skip connections) and training hyperparameters (e.g., batch size, learning rate, optimizer choice) can substantially affect the geometry of the loss surface. Building on these insights, we investigate how GRAFT-based subset training modifies the loss landscape relative to conventional full-data training. While a comprehensive theoretical analysis of data pruning and landscape geometry is beyond the present scope, we provide empirical evidence and defer in-depth investigation to future work.

**Experimental Setup.** We train ResNet-9 on CIFAR-10 with SGD (momentum 0.9, weight decay $5 \times 10^{-4}$, batch size 128, cosine-annealed learning rate) in two regimes, the full dataset and a GRAFT-selected 25% subset chosen by our reference GRAFT selector (per-batch SVD features, Fast MaxVol selection, candidate rank picked by the projection-residual criterion). Following the filter-normalized scheme of Li et al. (2018), we visualize each trained minimizer by drawing two Gaussian directions in weight space, normalizing each filter-wise to the magnitude of the corresponding weights, and evaluating the training loss on a $21 \times 21$ grid spanning $[-1, 1]^2$. Filter normalization makes the apparent sharpness comparable across models, and both contour panels share a single loss colour scale, so geometry differences are read directly rather than through axis rescaling.

Figure 6 compares the loss surfaces for models trained on the complete dataset versus those trained on a GRAFT-selected subset. In both regimes the trained minimizer sits at the centre of a smooth, predominantly convex basin, with no sharp ridges or irregular non-convex structure introduced by subset selection. Measured over the same filter-normalized window, the training loss varies by 2.92 for the full-data model and 1.43 for the GRAFT-subset model (shared colour scale in Fig. 6); if anything, the GRAFT-subset minimizer lies in a slightly *flatter* basin rather than a sharper one. This indicates that GRAFT preserves a favorable, well-localized loss geometry comparable to full-data training, consistent with its small accuracy gap at moderate fractions.

## 12 Details on Datasets

**CIFAR-10.** The CIFAR-10 dataset comprises 60,000 colorful images, each with dimensions of $32 \times 32$ pixels, distributed across 10 distinct categories. Within each category, there are precisely 6,000 images, contributing to a balanced distribution. This dataset is segregated into five training sets and one test set, with each set containing 10,000 images. Specifically, the test set is composed of precisely 1,000 randomly selected images from each category, ensuring representation across classes.

**CIFAR-100.** Similar to CIFAR-10, the CIFAR-100 dataset boasts 100 classes, each featuring 600 images. Within each class, there are 500 training images and 100 testing images, maintaining a balanced distribution for robust model training and evaluation. However, CIFAR-100 introduces a hierarchical structure by grouping

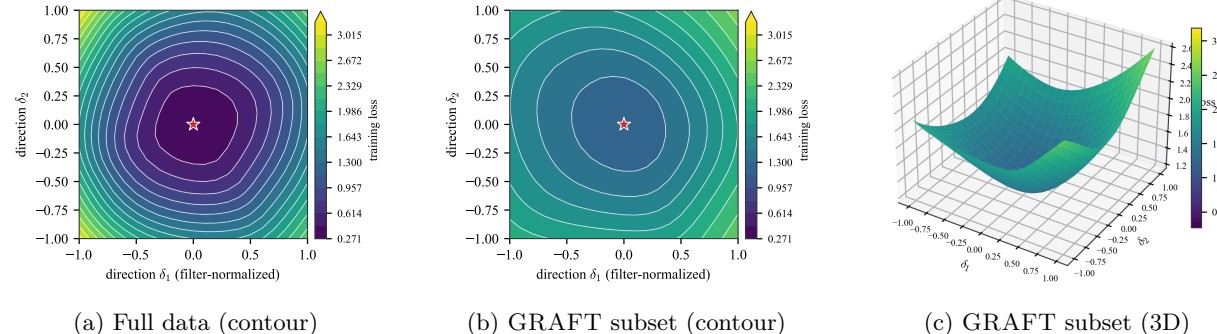

(a) Full data (contour)      (b) GRAFT subset (contour)      (c) GRAFT subset (3D)

Figure 6: Effect of GRAFT subset training on the loss landscape (ResNet-9, CIFAR-10; filter-normalized loss surfaces following Li et al. (2018), shared loss colour scale; red ⋆ marks the trained minimizer), showing filter-normalized 2D contours for (a) the full-data and (b) the GRAFT-subset (25%) model, and (c) a 3D rendering of the GRAFT-subset loss surface. The minimizers are comparably sharp and well-localized with predominantly convex basins, indicating that GRAFT selection only mildly perturbs the loss geometry.

the 100 classes into 20 superclasses. Every image in this dataset is assigned two labels, a "fine" label indicating its specific class and a "coarse" label denoting its superclass.

**Fashion-MNIST.** Fashion-MNIST, an image dataset by Zalando, comprises 60,000 training examples and 10,000 test examples, each depicting grayscale images sized 28x28. These images are classified into 10 distinct categories. Zalando's aim is to offer Fashion-MNIST as a direct substitute for the original MNIST dataset, enabling seamless benchmarking of machine learning algorithms. Notably, Fashion-MNIST maintains the same image dimensions and split structure as MNIST, facilitating effortless comparisons between models.

**Dermamnist.** The Dermamnist dataset is part of the MedMNIST collection and is designed for the classification of skin lesion images. It contains a diverse set of images representing various skin conditions, annotated with corresponding labels for seven different dermatological diseases. The dataset is intended for use in medical image analysis and machine learning research, providing a benchmark for evaluating models in the context of dermatology.

## 13 Feature selections models

In this section we introduce different feature selection methods such as SVD, ICA, PCA and Convolutional Neural Network (CNN) based Encoders.

- **Singular Value Decomposition (SVD)** Singular value decomposition (SVD) is a method of representing a matrix as a series of linear approximations that expose the underlying meaning-structure of the matrix. The goal of SVD is to find the optimal set of factors that best predict the outcome. SVD has been used to find the underlying meaning of terms in various documents. SVD reduces the overall dimensionality of the input matrix to a lower dimensional space (a matrix of much smaller size with fewer variables), where each consecutive dimension represents the largest degree of variability (between terms and documents) possible Miner et al. (2012).

- **Independent Component Analysis (ICA)** Independent Component Analysis (ICA) is a statistical method used to identify hidden factors of random variables. It is a linear generative model which assumes the observed variables are a linear mixture of unknown non Gaussian and mutually independent variables. The aim of ICA is to find those variables without making any assumptions about the mixing system. More formally, if the data are represented by the vector $x = (x_1, \ldots, x_n)$ and the independent component by the vector $= (s_1, \ldots, s_n)$, the aim of ICA is to find a linear transformation

W verifying s = Wx and minimizing a function F measuring the statistical independence Hyvärinen & Oja (2000)

- **Principle Component Analysis PCA** Principal component analysis (PCA) is one of the most widely used multivariate analysis techniques. The aim of PCA is to reduce the data to a few characteristic dimensions for visualisation and analysis. This is achieved by calculating a new set of variables, or principal components, each of which is a linear combination of the original variables. The principal components (PCs) are chosen so that the important information in the data is retained in just a few of these new variables, effectively summarising the samples or observations. Better approximations are obtained by using more PCs, where each successive PC is uncorrelated with the previous PCs and expresses as much of the remaining variance as possible. This is achieved from the eigenvalue decomposition of the data covariance matrix with the coefficients for the k'th principal component determined by the eigenvector of the covariance matrix corresponding to the kth largest eigenvalue. Data reduction is achieved by only keeping the first few PCs, which contain most of the information in the data Jolliffe (2002).

- **CNN based Encoders** Convolutional autoencoder extends the basic structure of the simple autoencoder by replacing the fully connected layers to convolution layers. Encoder use convolutions and decoders use transposed convolutional layers. CNN based encoders can reduce the less important feature coefficients to zero and the magnitude of the coefficients can be used as feature scores.

