# OpenReview forum: "GRAFT: Gradient-Aware Fast MaxVol Technique for Dynamic Data Sampling"
_TMLR — Under review for TMLR_

### Review · Reviewer_1RpJ · 2026-07-10

**Summary Of Contributions:**

This paper proposes GRAFT, a data subset selection method for efficient training. GRAFT constructs a low-rank feature representation within each mini-batch, uses Fast MaxVol to select representative examples, and adjusts the rank/subset size using the projection residual of the full current mini-batch gradient. The central claim is that this geometry-driven selection, with gradients used mainly to calibrate the budget, reduces training time, energy use, and $\mathrm{CO}_2$ emissions while preserving accuracy.

Key strengths:

1. The problem is important and practically relevant for data-efficient and sustainable training.
2. The method is relatively simple and distinct from direct gradient matching as in GradMatch and, more broadly, from integrated gradient-based subset-selection methods such as GLISTER.
3. The experiments cover several vision datasets, ImageNet-1K, BERT/IMDB, and an instruction embedding diversity study.
4. The paper reports $\mathrm{CO}_2$ emissions and discusses several regimes where the method does or does not help.

Key weaknesses:

1. The theoretical analysis does not fully match the apparent subset-gradient update used in training.
2. The dynamic-rank rule is not consistently defined across the motivation, equations, and pseudocode.
3. Dominant refresh/gradient costs and the data-budget accounting are unclear.
4. The main results are single-run, and the supplement contains configuration and $\mathrm{CO}_2$ inconsistencies.
5. The instruction-data curation part demonstrates embedding-space diversity rather than downstream instruction-tuning performance.

**Additional Comments:**

The main issues are not limited benchmark breadth or minor missing ablations. They concern the alignment of the paper’s central components: feature-space MaxVol selection, gradient-space rank checking, projected-gradient theory, and apparent unweighted subset-gradient training. Together with unclear dynamic-rank/cost definitions, single-run main results, and unresolved configuration/emissions inconsistencies, these issues prevent the current evidence from supporting the paper’s central claims.

**Audience:**

Yes

**Audience Explanation:**

Yes. Researchers interested in data-efficient training, coreset/subset selection, sustainable machine learning, and training-energy evaluation would likely find the findings relevant. GRAFT offers a route different from direct gradient matching, and the reported accuracy-emissions trade-offs are potentially useful, especially in aggressive-pruning and ImageNet-1K settings. The paper also reports boundary cases, such as cold GRAFT being statistically indistinguishable from random sampling on CIFAR-10.

**Broader Impact Concerns:**

The current submission does not present an apparent major ethical concern. Since the paper makes environmental claims, it should report the actual carbon intensity/region and specify whether warm-start, data loading, and selection overheads are included. The DermaMNIST result should not be interpreted as clinical evidence, and any expanded LLM data-selection claims should discuss data bias, coverage gaps, and safety implications.

**Claims And Evidence:**

No

**Claims Explanation:**

The current version does not support several central claims with sufficiently accurate, convincing, and clear evidence. The main issues concern the actual update rule, the dynamic-rank algorithm, computational accounting, and the robustness/reproducibility of the experiments.

First, there is a gap between the theoretical result and the actual algorithm. The convergence analysis is mainly about an idealized projected full current mini-batch gradient direction, while the implemented training appears to update on the selected subset, closer to an unweighted subset mean gradient. These two directions are not generally equivalent. The projection residual is therefore not an upper bound on the actual subset-update error, and Remark 1 already notes that MaxVol subspace coverage does not imply a closed-form bound between the subset mean gradient and the full current mini-batch mean gradient. Thus, the convergence result applies to an idealized projected direction, not to the apparent implemented GRAFT update.

Even for the idealized projected-gradient update, the appendix appears to overstate and inconsistently formulate what is proved. The displayed derivations provide an averaged or minimum-iterate bound on the squared gradient norm, whereas the text states convergence of the iterates to a stationary point. The dependence on the projection tolerance is also not reconciled across the theorem statements and proofs, which variously involve terms of order $\varepsilon G$ and $\mathcal{O}(\varepsilon^2)$. These statements should be made mathematically consistent.

Second, the dynamic-rank rule is not precisely defined. The text motivates selecting the smallest rank satisfying an error tolerance, but the equations and Algorithms 1/2 read more like selecting the rank with the smallest residual, or minimizing the residual among feasible ranks. These are different rules and can favor larger ranks. The no-feasible-rank case and tie handling are unspecified. The paper should also state whether MaxVol is rerun independently for each candidate rank and whether the resulting projection residuals are expected to be monotone or otherwise comparable across ranks. The main tables report fixed fractions, but the paper does not show rank trajectories, average selected fractions, rank distributions, or the fraction of steps satisfying the tolerance.

Third, the complexity and training procedure require clarification. Algorithms 1/2 describe a refresh loop over training mini-batches, computing the full current mini-batch gradient and selected per-sample gradients. If this is a full refresh pass, the cost scales with the number of batches; if it is only the current optimizer batch, the pseudocode and aggregation of $S_t$ need revision. If full-parameter per-sample gradients are used, the projection check requires $G_R\in\mathbb{R}^{|\Theta|\times R}$; for a 92M-parameter BERT model and $R=10$, this is already about 3.7 GB in fp32 before other overheads. The paper should specify whether it uses full gradients, last-layer gradients, gradient embeddings, blockwise projection, or another approximation. The notation also uses $d$ ambiguously as both feature dimension and gradient-matrix row dimension, although a full parameter gradient has dimension $|\Theta|$.

Fourth, the empirical support is not robust enough. The main per-dataset tables are single-run, while the only multi-seed analysis is on CIFAR-10 with a simplified pipeline rather than the main CORDS setup. In that analysis, GRAFT is not statistically distinguishable from random sampling at any fraction. The strongest claims, however, rely on CIFAR-100, TinyImageNet, Caltech256, and ImageNet-1K aggressive-pruning regimes, where comparable uncertainty estimates are not provided.

Finally, the reporting has reproducibility gaps. The supplement lists multiple architectures, batch sizes, and refresh schedules without clearly mapping each table to a configuration. Table 6 and Table 13 also disagree on CIFAR-10 GRAFT Warm at 5% $\mathrm{CO}_2$ (0.0480 vs. 0.0280) with the same accuracy. The actual carbon intensity, warm-start accounting boundary, wall-clock/energy values, and the meaning of “using only 30% of the data” under dynamic refreshes are also unclear.

**Requested Changes:**

Requested revisions:

The following items specify the changes required to make the algorithm, theoretical scope, computational cost, and empirical evidence sufficiently clear and reproducible.

1. Clarify the theory/update mismatch. The paper should state whether the Theorem/Corollary cover only the projected full current mini-batch gradient or also the unweighted subset gradient used in training. If the latter, an additional proof is needed; if not, statements such as “GRAFT ensures convergence” should be limited to the idealized projected update. The appendix theorem should also be reconciled with the averaged/minimum-iterate squared-gradient bounds actually derived and with the inconsistent tolerance dependences appearing across statements and proofs.

2. Correct and specify the dynamic-rank rule. The motivation, equations, and Algorithms 1/2 should be made consistent: does the method select the rank with the smallest residual, or the smallest rank satisfying the $\varepsilon$ tolerance? The pseudocode should cover the no-feasible-rank case, tie handling, whether MaxVol is rerun independently for each candidate rank, and whether residuals are expected to be monotone or otherwise comparable across ranks. The paper should also report rank trajectories, average selected fractions, rank distributions, and the fraction of steps satisfying the tolerance.

3. Clarify algorithmic cost and data budget. The paper should distinguish per-batch selection cost, total refresh cost, complete training cost, and wall-clock/$\mathrm{CO}_2$ cost. It should specify how the full current mini-batch gradient reference is computed, whether each refresh traverses all batches, and how $G_R\in\mathbb{R}^{|\Theta|\times R}$ is implemented if full-parameter per-sample gradients are used. The ambiguity in $d$ as both feature dimension and gradient-matrix row dimension should also be resolved. For claims such as “using only 30% of the data,” the paper should report per-step selected fraction, cumulative unique-sample coverage, total forward/backward sample evaluations, and the random baseline refresh policy.

4. Strengthen empirical support and reproducibility. The main accuracy claims should be supported by multi-seed results on at least one or two key settings, such as CIFAR-100/TinyImageNet aggressive pruning, with mean/std or confidence intervals and clear comparisons to random/GradMatch/GLISTER. The paper should also provide a per-table mapping of architecture, batch size, epochs, optimizer, learning rate, candidate ranks, threshold, refresh interval, warm-start length, hardware, and carbon intensity.

5. Resolve emissions and configuration inconsistencies. The CIFAR-10 GRAFT Warm 5% $\mathrm{CO}_2$ discrepancy between Table 6 and Table 13 should be corrected or explained. The paper should specify the full-data warm-up length for GRAFT Warm, whether the warm-up and selection overheads are fully included in $\mathrm{CO}_2$, and provide the wall-clock and energy values claimed in the text, not only emissions.

Additional clarifications and improvements:

1. Figure 3 should separate measured operating points from monotone non-decreasing interpolation. The monotonicity assumption is contradicted by the paper’s own FashionMNIST GRAFT Warm results, which decrease from 92.95% at 25% to 92.62% at 35%.

2. The relationship between the sequential Fast MaxVol heuristic and standard MaxVol/D-optimal design should be clarified. In particular, the basis-orientation dependence and the reduced-subspace nature of the D-optimality claim should be made explicit.

3. Several secondary claims should be tightened or completed: “gradient-aware” should be named more precisely, loss-landscape visualization should not be described as convexity analysis, and the instruction-data curation section should remain clearly preliminary unless downstream instruction-tuning results and relevant baselines are added.

---

### Review · Reviewer_joHN · 2026-07-11

**Summary Of Contributions:**

This paper proposes a method called GRAFT, for data subset selection in machine
learning. The method aims to represent the full data set with a smaller subset.
It uses SVD to find a low-rank representation of the batch and then adaptively
modifies the rank by comparing the residual of the full gradient vector and the
projected gradient vector. The authors report that it can, for many data sets,
retain good accuracy while leaving a smaller carbon footprint and using less
training time.

The key strengths of the paper are that its results are promising, the method is
both well-motivated and simple enough to be implemented, and the empirical
validation is quite extensive.

The weaknesses of the paper are that there are comparator methods that are not
included in the experiments, sometimes without justification, that the code is
not available, and that some of the theoretical claims are loose, contradictory,
or not aligned with the algorithm as presented.

**Additional Comments:**

- In Table 2, the ICA probe accuracy is $38.25 \pm 0.00\%$ over five seeds. It's
  surprising that the standard deviation is exactly zero (or even rounded to
  that). This is worth double-checking.

**Audience:**

Yes

**Audience Explanation:**

The paper is well-written and concerns a topic of interest to a wide audience of
both researchers and practitioners using deep learning.

**Broader Impact Concerns:**

No, I don't think there are any broader impact concerns here

**Claims And Evidence:**

No

**Claims Explanation:**

The experiments are convincing, quite comprehensive, and well-presented. It is
also refreshing to see the honest reporting of the null result on CIFAR-10.
There are, however, issues that make it difficult to fully validate that the
claims are supported by the evidence:

- There is no code available for the paper. The paper states that it will be
  released in the camera-ready version, but this is a paper that is primarily
  empirical and it is currently impossible to validate the claims without access
  to the code.
- The paper repeatedly states that Algorithm 1 has complexity independent of the
  dataset size, yet the algorithm as written loops over all batches in the
  dataset at each refresh step. This is not independent of dataset size, and it
  is not clear whether the algorithm box misrepresents the implementation or
  whether the complexity claim is incorrect.
- The multi-seed experiment of section 10 shows that GRAFT is approximately
  equal to the random baseline on CIFAR-10, yet the random baseline is not
  included in the other experiments (Tables 6-9, 11-12). Why was it omitted
  here? This begs the question whether the claim that GRAFT is better than
  random on harder datasets is actually supported by the evidence.
- The one harder-dataset random comparison that exists (Table 10, ImageNet-1K)
  reports random sampling at 21.12% top-1 at $f = 0.05$ (33.51% at $f = 0.10$,
  55.12% at $f = 0.30$). These seem to be low. Under a closely matched
  protocol (ImageNet-1K, ResNet-18, fixed subsets, comparable full-data
  accuracy of 69.52% vs. the paper's 70.36%), the DeepCore benchmark that the
  paper cites (Guo et al., 2022, Table 2) measures random selection at
  $40.09 \pm 0.21$% at 5%, $52.10 \pm 0.22$% at 10%, and $64.11 \pm 0.05$% at
  30%, i.e., 9-19 points above the values in Table 10, and finds GradMatch
  approximately at or below random at every fraction, versus the 26-point
  GradMatch advantage over random implied by Table 10. (I note that DeepCore
  evaluates GradMatch in a slightly different estting, but it cannot explain the random
  baseline differing.) Please specify the random-baseline protocol precisely
  (fixed vs. resampled subset; number of epochs; whether epoch-, step-, or
  compute-matched to GRAFT) and confirm that $f$ denotes the same budget for
  random as for GRAFT. Since GRAFT selects per batch from the full dataloader,
  the union of samples it trains on across refreshes may exceed $fn$, whereas a
  fixed random subset is capped at $fn$ distinct samples.
- CRAIG is mentioned exactly once (in related works) and not included in the
  experiments. It's not clear why it was omitted, and it is a relevant
  comparator method. The same applies to BADGE, SelMatch, and SubSelNet. The
  paper should either include these methods in the experiments or justify their
  omission.
- Assumption 3 (A3) requires that the norm of the difference of the gradient and
  projected gradient is bounded at *every step* by $\varepsilon$, yet the
  algorithm only checks this condition at each refresh step. This is not
  directly examined in the paper as far as I can tell, and it is not clear that
  the assumption is satisfied in practice.
- Theorem 2 has some problems:
  - The inner product $\langle \nabla L, g_\text{proj} \rangle$ is bounded with a
    Cauchy-Schwarz inequality based on the *full* gradient, but only a
    mini-batch version thereof is available.
  - The paper concludes $\lVert \nabla L(\Theta_t) \rVert \leq \varepsilon G$, but that
    does not follow from the derivation. The telescoped inequality bounds the
    *average of the squared* gradient norms by $\varepsilon G + o(1)$, so the
    correct conclusion is
    $\min_{t \le T} \lVert \nabla L(\Theta_t) \rVert \leq \sqrt{\varepsilon G} + o(1)$:
    there is a missing square root, and the bound holds for the best iterate (or
    on average), not for every iterate as the phrase "converges to a stationary
    point where..." suggests.
  - It is referenced once as yielding a "stationary neighborhood" and once as
    yielding a "stationary point". These are not the same thing.
- $\varepsilon$ is never actually given a value in the experiments. They are
  instead parameterized (reported) in terms of relative residual, but how this
  related to $\varepsilon$ is not clear. And there are no guidelines for how to
  choose $\varepsilon$ in practice, which makes the method less useful for
  practitioners.
- The entire pre-selection family of methods is ignored in the experiments, even
  though they are recent and relevant (particularly with respect to the focus on
  carbon footprint).
- Several internal inconsistencies make it difficult to establish which
  experimental setup actually produced the headline results:
  - Section 9.2 states that "unless noted otherwise" runs use ResNet-18 with
    batch size 128 and that every figure is produced from these measured
    quantities, while section 9.4 attributes the main-table experiments
    (Tables 6, 7, 11) to ResNeXt29_32x4d with batch size 200. The main text
    (section 4.1, Fig. 3) never states the architecture. Moreover, Appendix 10
    acknowledges two distinct pipelines ("the simplified reference pipeline,
    not the CORDS setup used for the main tables"), and Figure 2, the
    evidence for GRAFT's selection mechanism, is measured on the reference
    pipeline (ResNet-18, $K{=}128$), i.e., a different configuration from the
    one behind the headline tables. Please state explicitly, per table and
    figure, which pipeline, architecture, and batch size was used.
  - Section 3.3 recommends batch sizes of at least 128, yet per section 9.4
    TinyImageNet and Caltech256 were run at batch size 100, below the paper's
    own recommended operating regime.
  - Table 6 and Table 13 report the same GRAFT Warm runs (all other cells are
    identical), but at $f=0.05$ the emissions differ: 0.0480 kg vs.
    0.0280 kg at identical accuracy (90.86%). At least one is erroneous, and
    this cell lies exactly in the $\mathcal{E} < 0.05$ kg region where the
    warm-start claim of section 4.1 is made.
  - GRAFT reports identical accuracy (40.6%) at $f=0.05$ on both CIFAR-100
    and TinyImageNet (Tables 7-8, repeated in the section 4.1 prose) despite
    very different full-data accuracies (75.45% vs. 59.0%), and identical
    emissions (0.049 kg) at $f{=}0.05$ on TinyImageNet and Caltech256
    (Tables 8-9). Possibly coincidence, but can you please verify these against the
    raw logs?
  - Some emissions are non-monotone in the data fraction under a fixed-epoch
    protocol: GLISTER on CIFAR-10 drops from 0.0938 kg ($f=0.25$) to
    0.0846 kg ($f=0.35$) (Table 6), and DRoP on TinyImageNet costs less at
    15% than at 5% (Table 8). Can you please verify these claims?

Most of these issues are correctable, but they do make it difficult to validate the
claims of the paper as currently written.

**Requested Changes:**

### Critical

- Theorem 1 and 2 yield neighborhoods of different order. Corollary 1 cites
  Theorem 2 while the main paper cites Theorem 1. The paper should clarify and
  reconcile these differences.
- Clarify the complexity claim of Algorithm 1, and whether it is independent of
  dataset size. If it is not, please correct the claim.
- Release the code for the paper now, so that the results can be validated.
- Include the random baseline in all experiments, or justify its omission.
- Clarify how $\varepsilon$ is chosen in practice, and how it relates to the
  reported relative residual.
- Clarify how Assumption 3 is satisfied in practice, given that it is only
  checked at the end of each epoch.
- In Preliminaries (section 2), the paper asserts
  $\lVert \bar{g} - \tilde{g} \rVert_2 \le \varepsilon$ for the *unweighted
  subset mean* $\tilde{g}$, presenting it as a property of the method. However,
  no mechanism in the paper enforces this inequality: the dynamic-rank rule
  controls the projection residual $\lVert \bar{g} - P_R \bar{g} \rVert_2$ (the
  *optimal* linear combination of the selected gradients, cf. Lemma 1), which
  does not bound the deviation of the fixed unweighted combination $\tilde{g}$.
  The paper itself acknowledges this gap: section 3.2 states that the
  subset-mean deviation is tracked empirically "rather than bound it in closed
  form", and Remark 1 shows that no $\sigma_{R+1}$-based bound on it can exist.
  Two different error quantities are conflated in the setup. Please restate the
  Preliminaries in terms of the quantity that is actually controlled, or justify
  the asserted inequality.
- Theorem 2, Lemma 1, Corollary 1 are each stated *twice* in the appendix with
  slightly different wording ("stationary neighborhood" vs. "stationary point").

### Minor (Typos, grammar, notation)

- Include CRAIG, BADGE, SelMatch, and SubSelNet in the experiments, or justify
  their omission.
- Refresh-interval units are inconsistent: $S = 20-50$ *iterations* (section
  3.2), $40-50$ (section 3.3), $S \in [20, 50]$ (section 9.2), but Figure 2's
  caption says "refreshed every five *epochs*" and section 9.3 says "every 10
  *epochs*."
- The text (axis labels etc) in Figure 3 is too small. Please enlarge it so that
  it is legible.
- "MaxVol" is spelled at least five ways: *MaxVol*, *Maxvol*, *Max-volume*,
  `fast-maxvol` (Algorithm 1), `fastmax-volume` (Algorithm 2, Table 3). Pick
  one.
- Mixed British/American spelling: "maximisation/normalising/behaviour"
  (sections 2, 4.2) vs. "maximization/behavior" (sections 1 and 11).
- "ResNet18" vs. "ResNet-18" hyphenation varies (e.g., section 3.2 vs.
  abstract).
- Section 13 heading: "Feature select**ions** models" should be "Feature
  selection models"; same section: "**Principle** Component Analysis" should be
  "Principal"; ICA bullet: "the independent component by the vector
  $= (s_1, \ldots, s_n)$" is missing the symbol $s$ before "=".
- Feature dimension has three different notations: $M$ (section 3.1,
  $A \in \mathbb{R}^{K \times M}$), $m$ (section 3.2, "batches of size
  $m \times K$", $m$ never defined), and $d$ (notation table). Meanwhile $d$
  also serves as parameter dimension: preliminaries put per-sample gradients in
  $G \in \mathbb{R}^{d \times K}$, but gradients are in $\mathbb{R}^{|\Theta|}$;
  the notation table itself says $g_r \in \mathbb{R}^{|\Theta|}$ while
  $G \in \mathbb{R}^{d \times K}$.
- $\epsilon$ and $\varepsilon$ are used interchangeably for the same tolerance
  (section 3.2 vs. section 3.3, Theorem 1). The bound alternates between
  $\|\cdot\|_2 \le \epsilon$ and $\|\cdot\|_2^2 \le \varepsilon$ across section
  3.2, (A3), Lemma 1, Corollary 1.
- Equation numbering restarts: (1)-(3) in section 9.1 collide with Eq. (1) in
  the main text and Eqs. (3)--(4) in section 9.4.
- Some references are wrong in the bibliography, check the title for Bottou for
  instance. There's a duplicated entry for Krizhevsky and Hinton. There are also
  some mistakes like "AndreiChertkov".
- In section 4.2, the paper writes "maximises $\det(\Phi_S^\top \Phi_S)$", but for
  $R < d$ this $d \times d$ matrix is rank-deficient and the determinant is
  zero. Figure 4 correctly uses $\det(\Phi_S \Phi_S^\top)$. Text should match, I
  think.

---

### Review · Reviewer_9Nb8 · 2026-07-17

**Summary Of Contributions:**

The paper proposes GRAFT, a gradient-aware technique for dynamic data sampling, and dynamically adjusts the subset sizes using a gradient-approximation criterion. The method is evaluated across various benchmarks and shows good results.
**strengths:**
- The method is conceptually clear. Using low-rank features plus MaxVol sampling is a neat alternative to gradient-matching methods such as GradMatch and GLISTER.
- The experiments are broad, covering several vision datasets, ImageNet-1K, BERT fine-tuning, and preliminary instruction-data curation.

**weaknesses**
- The theoretical analysis is somewhat limited. The connection between MaxVol-selected samples and accurate full-batch gradient approximation is not fully proven for nonlinear neural networks.
- The warm-start version improves results, but it uses extra full-data warm-up, so it should not be compared too directly with cold-start baselines.
- The instruction-data curation experiment only evaluates embedding-space diversity, not downstream LLM fine-tuning performance.
- More related methods should be discussed in the Related Work section, e.g., [1-3]

[1] Infobatch: Lossless training speed up by unbiased dynamic data pruning.

[2] RL-Selector: Reinforcement Learning-Guided Data Selection via Redundancy Assessment

**Audience:**

Yes

**Audience Explanation:**

See comments above.

**Claims And Evidence:**

No

**Claims Explanation:**

See comments above.

**Requested Changes:**

See comments above.